# Dual asparagine-depriving nanoparticles against solid tumors

Yubo Shen[1,3], Huifang Wang[1,3], Daoxia Guo[1], Jiantao Liu[2], Jinli Sun[1], Nan Chen[2] ✉, Haiyun Song ●[1] ✉ & Xiaoyuan Ji ●[1] ✉

Depletion of circulatory asparagine (Asn) by L-asparaginase (ASNase) has been used for clinical treatment of leukemia, whereas solid tumors are unresponsive to this therapy owing to their active Asn biosynthesis. Herein, we develop a type of core-shell structured cascade-responsive nanoparticles (NPs) for sequential modulation of exogenous Asn supply and endogenous Asn production. The reactive oxygen species-sensitive NP shells disintegrate in the tumor microenvironment and liberate ASNase to scavenge extracellular Asn. The acid-labile NP cores subsequently decompose in the tumor cells and release rotenone to block intracellular Asn biosynthesis. Administration of the dual Asn-depriving NPs in murine models of triple-negative breast cancer and colorectal cancer substantially suppress the growth and epithelial-mesenchymal transition of primary and relapsed tumors, fully eradicate spontaneous and post-surgical metastasis, and confer long-term T cell memory for complete resistance to tumor rechallenge. This study represents a generalized strategy to harness amino acid depletion therapy against solid tumors.

Increased nutrient uptake and enhanced anabolic metabolism are critical for rapid cancer cell proliferation and metastasis[1–3]. Amino acids are key nutrients and also serve as signaling molecules for metabolic and transcriptional regulation. Several types of amino acids have shown particular importance in promoting tumor growth or immune evasion[4–6]. For example, tumor cells are more strongly addicted to glutamine than to any other amino acid, which fuels the tricarboxylic acid (TCA) cycle, nucleotide and fatty acid biosynthesis, and redox balance, thereby supporting biosynthetic and pro-growth phenotypes[7,8]. As a main source of methyl donor, methionine metabolism is essential for epigenetic remodeling that drives cancer initiation, progression and therapeutic resistance[9,10]. Furthermore, tumor cells and tumor-infiltrating immune cells, such as dendritic cells (DCs) and macrophages, overexpress indoleamine 2,3-dioxygenase (IDO) for tryptophan metabolism and lead to kynurenine accumulation in the tumor microenvironment (TME), impeding T-cell activation

and promoting immune escape[11,12]. These findings have sparked widespread interest in targeting specific amino acids as an option for cancer treatment.

Asparagine (Asn) is a well-documented target for metabolic interventions in cancer[13–15]. Intracellular Asn functions as an amino acid exchange factor for the uptake of extracellular serine, arginine, and histidine[16]. In addition to providing building blocks for protein synthesis, Asn exerts pleiotropic effects on the anabolic pathways of nucleic acids, lipids and other macromolecules[16–18]. Recent studies have also revealed a strong correlation between Asn bioavailability and the metastatic potential of solid tumors[19,20]. In addition, systemic modulation of circulatory Asn has complex impacts on T cells, including their activation, their anti-tumor responses, and their ability to differentiate into memory cell phenotypes[21–23]. Moreover, the Asn-hydrolyzing enzyme L-asparaginase (ASNase) has been used clinically for the treatment of hematological malignancies, which heavily rely on

[1]School of Public Health, Shanghai Jiao Tong University School of Medicine, Shanghai, China. [2]College of Chemistry and Materials Science, The Education Ministry Key Lab of Resource Chemistry, Joint International Research Laboratory of Resource Chemistry of Ministry of Education, Shanghai Key Laboratory of Rare Earth Functional Materials, and Shanghai Frontiers Science Center of Biomimetic Catalysis, Shanghai Normal University, Shanghai, China. [3]These authors contributed equally: Yubo Shen, Huifang Wang. ✉e-mail: nchen@shnu.edu.cn; songhaiyun@shsmu.edu.cn; xyji@shsmu.edu.cn

exogenous sources of Asn for survival due to the silencing of endogenous biosynthesis[24–26]. However, ASNase treatment has so far failed to achieve successful therapeutic outcomes in solid tumors, and many challenges remain.

The metabolic flexibility of solid tumors allows them to adapt to nutrient-deprived conditions in the TME and renders inefficacy of extracellular-acting ASNase[27–29]. Most solid tumors can activate the expression of *asparagine synthetase* (*ASNS*) through a stress response effector activating transcription factor 4 (ATF4) and employ de novo biosynthesis as an alternative source of Asn, which maintains the intracellular Asn levels to support tumor cell proliferation in the absence of exogenous Asn and thereby abrogate the effectiveness of ASNase treatment[30–32]. In addition, it is a major challenge to achieve efficient local Asn restriction in the TME without causing severe systemic depletion. Therefore, strategies that precisely and concurrently target both extracellular Asn hydrolysis and intracellular Asn biosynthesis have great potential in enhancing the efficacy of Asn depletion therapy in solid tumors. Nanoparticle (NP)-based drug delivery systems have shown promise in co-delivering different types of therapeutic agents for cancer combination therapy[33–36]. Considering the complexity and anomaly of tumor metabolism, it is highly desirable to develop dual- or multi-stimuli-responsive NPs that are able to sense the characteristics of extracellular and intracellular environment and accordingly release appropriate metabolic regulators, accomplishing optimal efficacy with less side effects[37–40].

Herein, we fabricate a type of core-shell structured and dual-responsive NPs for concurrent modulation of internal Asn generation and external Asn supply in solid tumors. The acid-labile micellar cores are loaded with rotenone (Rot), which inhibits de novo Asn synthesis initiated from the TCA cycle[41–43]. The reactive oxygen species (ROS)-sensitive shells, composed of ASNase for extracellular Asn scavenging, are formed via reversible cross-linking to the micelle surfaces. These NPs are designed to sequentially release ASNase for exogenous Asn hydrolyzation and Rot for Asn biosynthesis inhibition, thereby restricting Asn availability in tumor cells. In mouse models of solid tumors, including triple-negative breast cancer (TNBC) and colorectal cancer (CRC), administration of these dual Asn-depriving NPs not only efficiently restrains tumor growth but also potently suppresses the potential of tumor recurrence and metastasis. Furthermore, treatment with these NPs induces long-term T-cell memory, providing cured mice with persistent and full protection from tumor rechallenge. Our study demonstrates a two-pronged approach for spatial regulation of Asn metabolism and extends the clinical potential of amino acid depletion therapy against solid tumors.

## Results

### Fabrication and characterization of dual Asn-depriving NPs

The dual Asn-depriving NPs were comprised of pH-responsive micellar cores and ROS-responsive ASNase shells. The amphiphilic poly(ε-caprolactone)-hydrazone-poly(ethylene glycol) (PCL-Hyd-PEG) copolymers, partially modified with tumor-targeting ligand hyaluronic acid (HA) at the hydrophilic terminals, were mixed with Rot for self-assembly into micelles (R-MHAs). The acid-labile Hyd linkers could mediate copolymer breakage in the endosomes, facilitating Rot release in tumor cells. Next, ASNase was conjugated to the R-MHA surfaces via the bis-*N*-hydroxy succinimide (NHS) cross-linkers embedded with ROS-cleavable thioketal (TK) moieties (NHS-TK-NHS), forming core-shell structured NPs (R-MAHAs) (Fig. 1a). The outer shells of R-MAHAs decomposed in response to abundant ROS in the TME and released free ASNase to hydrolyze exogenous Asn. The exposed inner cores were subsequently internalized by tumor cells and disassembled in acidic endosomes, releasing Rot for the blockade of endogenous Asn production. When applied to ASNase-insensitive solid tumors, these dual Asn-depriving NPs achieved satisfactory efficacy in the suppression of tumor growth and prevention of tumor recurrence and metastasis (Fig. 1b).

Dynamic light scattering (DLS) analysis revealed that the micelles without HA modification had an average diameter of $103.3 \pm 1.7$ nm. While HA functionalization increased the particle size to $133.7 \pm 2.6$ nm, the loading of Rot had a minimal impact on the size of MHAs (Fig. 2a). The encapsulated Rot in R-MHAs was quantified by its characteristic absorption peak at 298 nm, achieving a maximal loading efficiency of $21.1 \pm 1.5\%$ (Supplementary Fig. 1). The formation of ASNase shells at varying ASNase to R-MHA mass ratios (ranging from 1:40 to 1:5) led to stepwise increases in the particle size. The R-MAHAs exhibited an average size of $184.6 \pm 3.0$ nm and $58.3 \pm 1.5\%$ of ASNase was conjugated to the micellar surfaces when ASNase was mixed with the R-MHAs at a mass ratio of 1:15 (Supplementary Fig. 2). These NPs were negatively charged and a representative form of R-MAHAs showed a zeta potential value of $-9.2 \pm 0.2$ mV (Supplementary Fig. 3). Different particle sizes between R-MHAs and R-MAHAs were also confirmed by negative-stain transmission electron microscopy (TEM) (Fig. 2b). The resulting R-MAHAs were stable in cell culture medium, as indicated by their consistent size, homogeneity and surface charge (Fig. 2c and Supplementary Fig. 4). To assess whether the outer shells were covalently attached to the micellar cores via NHS-TK-NHS, we utilized fluorescein isothiocyanate (FITC)-doped R-MHAs (R-M^FITC^HAs) and cyanine 5.5 (Cy5.5)-conjugated IgG (IgG^Cy5.5^) to assemble dual-labeled R-M^FITC^IgG^Cy5.5^HAs. Poor colocalization levels between R-M^FITC^HAs and IgG^Cy5.5^ were observed when they were physically mixed together. In contrast, modifying R-M^FITC^HAs with the NHS-TK-NHS linkers significantly enhanced the colocalization ratios between R-M^FITC^HAs and IgG^Cy5.5^, indicating that the ASNase shells were predominantly formed through cross-linking to the micelles (Fig. 2d and Supplementary Fig. 5).

Tiered incorporation of TK and Hyd bonds into the R-MAHAs enabled sequential disintegration of ASNase shells and Rot-loaded micelles in the TME and intracellular environment, respectively (Fig. 2e). To validate this concept, we first measured the ASNase releasing curves of R-MAHAs at 1 mM hydrogen peroxide ($H_2O_2$), which mimicked the ROS levels in the TME[44,45]. As a comparison, free ASNase molecules were scarcely detected when the R-MAHAs were incubated with phosphate-buffered saline (PBS). The presence of $H_2O_2$ significantly accelerated the rate of ASNase release, with over 80% free ASNase detected in the supernatant within 120 min (Fig. 2f). Consistently, DLS analysis revealed ROS-dependent size shrinkage of the R-MAHAs, suggesting the disassociation of ASNase from the micellar surfaces after incubation with $H_2O_2$ (Fig. 2g). Next, we analyzed the kinetics of Rot release in the presence or absence of $H_2O_2$ pretreatment. The amount of Rot released from $H_2O_2$ pre-treated R-MAHAs was kept at basal levels under near-neutral conditions (pH 6.7−7.2) and was significantly increased under the endosome-mimicking condition (pH 5.0), validating the structural integrity of micellar cores under oxidative stress as well as the acid-dependent micelle disassembly and Rot release (Fig. 2h). In comparison, the absence of $H_2O_2$ pre-treatment noticeably decelerated the rate of Rot release from the R-MAHAs at acidic pH, implying that the prior breakdown of ASNase shells was necessary for optimized Rot release in tumor cells (Fig. 2i). Collectively, these results exhibited the potential of R-MAHAs as a cascade-responsive nanoplatform for sequential delivery of ASNase and Asn biosynthesis inhibitors into the TME and tumor cells, respectively.

### Dual Asn deprivation in vitro

We investigated the in vitro effects of these Asn-depriving NPs in cultured cells. HA molecules are high-affinity ligands for CD44 receptors that are overexpressed on many types of tumor cells[46–48]. The expression level of *CD44* in 4T1 breast cancer cells was substantially higher than that in undifferentiated bone marrow-derived

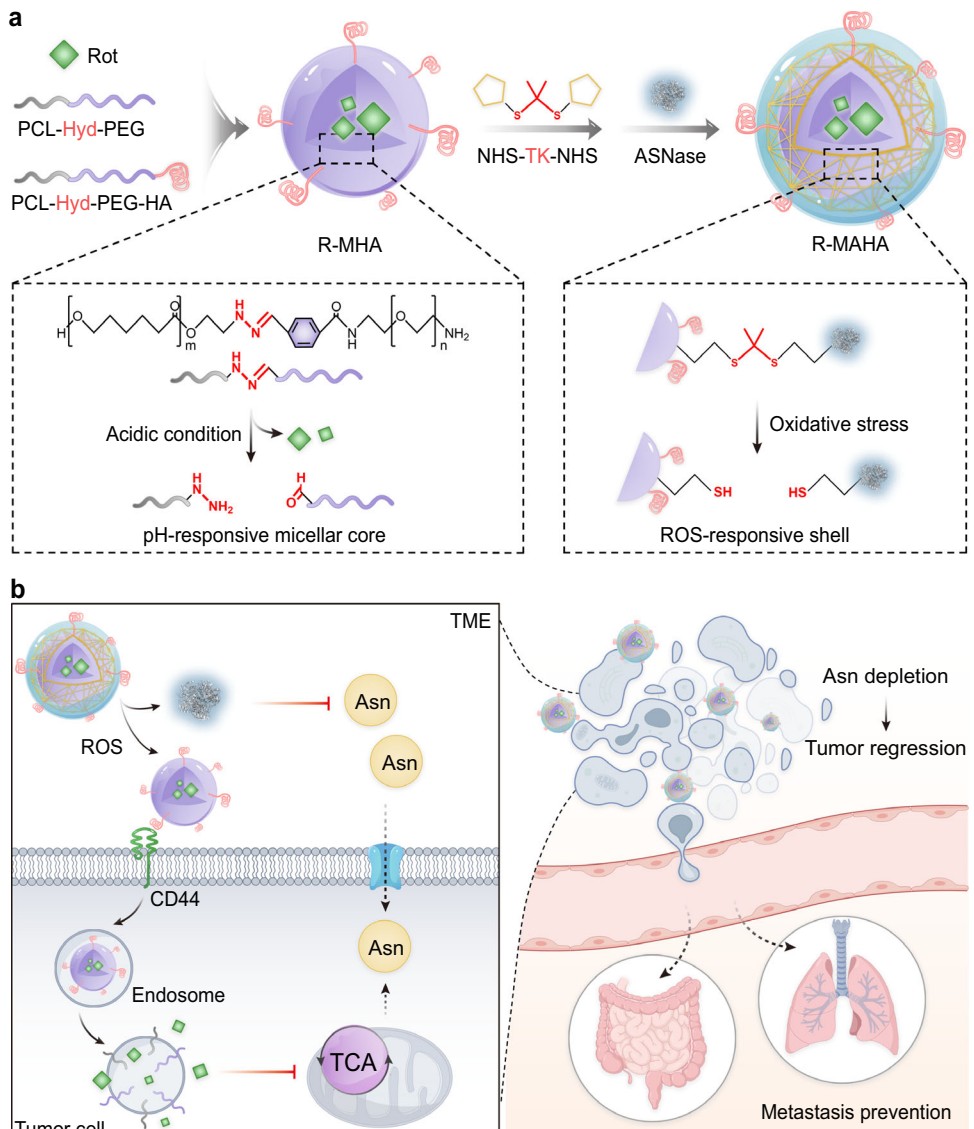

**Fig. 1 | Schematic illustration of two-pronged Asn modulation against solid tumors. a** A schematic representation of the construction of core-shell structured R-MAHAs and dual-responsive disintegration of Rot-loaded micellar cores and ASNase shells. **b** In the TME, R-MAHAs sequentially release ASNase for extracellular Asn hydrolysis and Rot for intracellular Asn biosynthesis inhibition. This dual modulation strategy synergistically restricts Asn availability in tumor cells, resulting in regression of tumor growth and prevention of post-surgical relapse and metastasis. Created in BioRender. shen, y. (2025) https://BioRender.com/f8okx09. The protein structure of ASNase was adapted from Cerofolini et al., with permission from John Wiley and Sons[58]. Rot rotenone, HA hyaluronic acid, NHS-TK-NHS *N*-hydroxy succinimide-thioketal-*N*-hydroxy succinimide, ASNase L-asparaginase, TME tumor microenvironment, ROS reactive oxygen species, Asn asparagine, TCA tricarboxylic acid.

macrophages (M0-BMDMs), naive CD8+ T cells, and activated CD8+ T cells (Supplementary Fig. 6). Compared to unmodified micelles, HA functionalization on the R-M^FITC HAs significantly enhanced cellular internalization of the micelles by 4T1 tumor cells rather than by M0-BMDMs or naive CD8+ T cells, and far more than by activated CD8+ T cells, validating the tumor cell-targeting capacity of the micellar surface-decorated HA (Fig. 3a and Supplementary Figs. 7 and 8). The internalized micelles were primarily localized in acidic organelles, allowing for the cleavage of Hyd linkers and the release of Rot (Fig. 3b and Supplementary Fig. 9). The cytotoxicity of R-MHAs was assessed in various types of cells. Unloaded MHAs did not compromise the viability of 4T1 breast cancer cells, Panc02 pancreatic cancer cells, LLC lung cancer cells, H22 hepatic cancer cells, M0-BMDMs, naive CD8+ T cells, or activated CD8+ T cells. The MHAs loaded with Rot (10−70 ng/mL) exhibited severe and dose-dependent inhibitory effects on the viability of 4T1, Panc02, LLC and H22 cells. In contrast, R-MHAs loaded

with 10−60 ng/mL Rot did not cause lethality in M0-BMDMs or naive CD8+ T cells, R-MHAs carrying 10−50 ng/mL Rot did not affect the viability of activated CD8+ T cells, and higher concentrations only mildly weakened the cell viabilities, manifesting that tumor cells were more sensitive to the inhibition of Asn synthesis than other cell types (Fig. 3c and Supplementary Fig. 10).

If the R-MHAs acted via suppressing Asn synthesis, the reduction of intracellular Asn levels would upregulate the transcription of metabolic genes involved in amino acid synthesis and transport, including *ASNS*, *PSAT1*, *GPT2*, *MTHFD2*, and *SLC1A5* (Fig. 3d)[30,49]. Accordingly, we monitored the expression levels of these critical genes after R-MHA treatment. The R-MHAs and an equivalent dose of free Rot promoted the expression of above genes to comparable levels, suggesting their similar efficiencies in inducing Asn starvation (Fig. 3e and Supplementary Fig. 11). Since solid tumors could maintain their intracellular Asn pool via both endogenous and exogenous sources, we

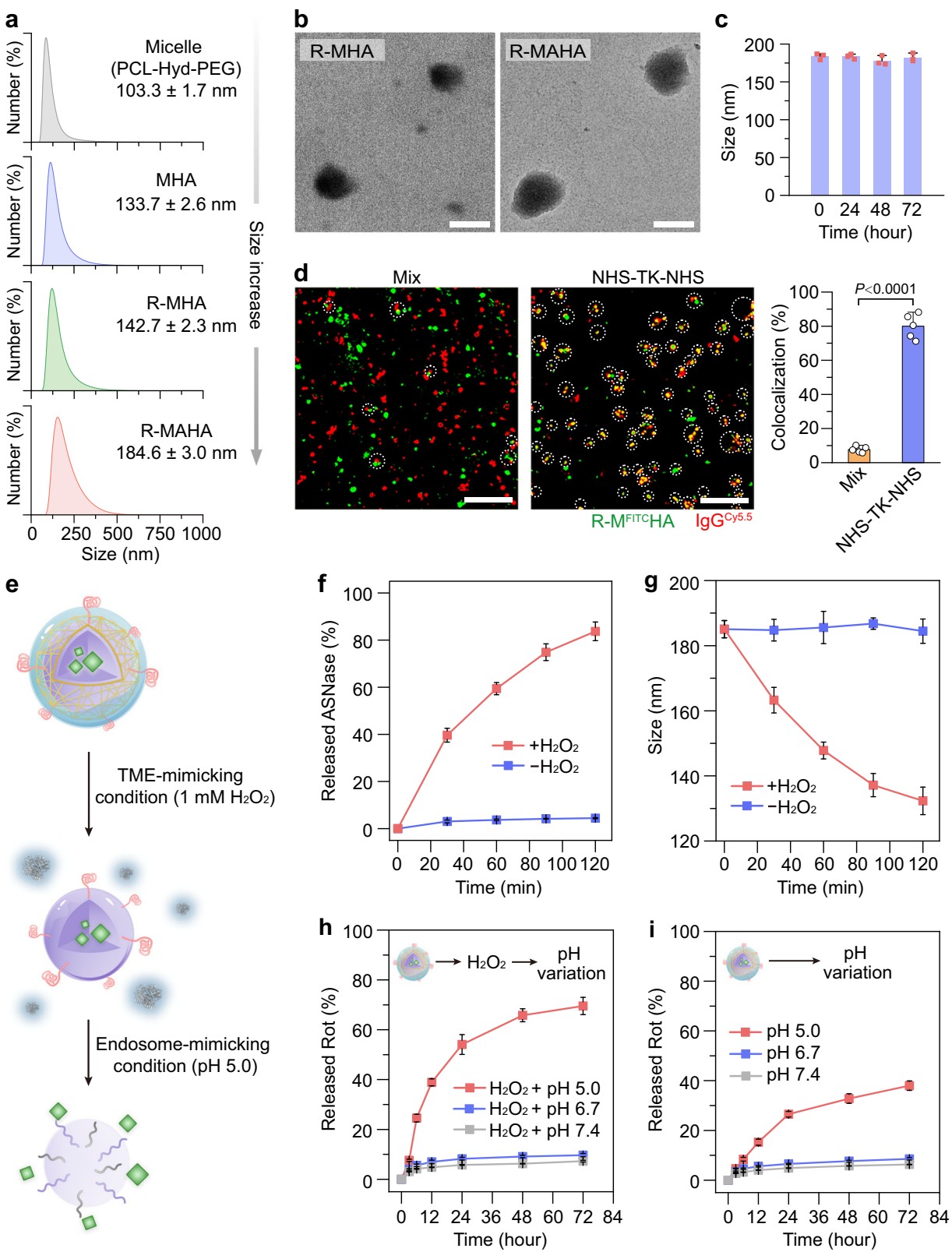

next evaluated the combined Asn-depriving effects by the R-MAHAs (Fig. 3f). The $H_2O_2$ pre-treated R-MAHAs and free ASNase exhibited very similar kinetics of Asn hydrolysis, indicating that the NP-released ASNase molecules retained their catalytic activities (Fig. 3g). Incubation with R-MHAs markedly reduced the intracellular levels of Asn in 4T1 tumor cells. The $H_2O_2$ pre-treated R-MAHAs exacerbated the degree of Asn deprivation, attributed to their additional ability in

scavenging Asn in the medium. Supplementing exogenous Asn (0.1 mM) in the cell culture medium could restore the intracellular Asn levels in the presence of R-MHAs, whereas this rescuing effect was abolished in the presence of $H_2O_2$ pre-treated R-MAHAs (Fig. 3h). In consistent with this observation, the R-MAHAs impaired the viability of 4T1 cells more severely than the R-MHAs both in the absence and presence of supplemented Asn (Fig. 3i). Similar effects were also

**Fig. 2 | Characterization of dual Asn-depriving NPs. a** Hydrodynamic diameters of indicated NPs. Data are represented as mean ± SD (*n* = 3 independent samples). **b** Representative TEM images of R-MHAs and R-MAHAs. Scale bars: 100 nm. **c** Hydrodynamic diameters of R-MAHAs dispersed in cell culture medium at various time points. Data are represented as mean ± SD (*n* = 3 independent samples). **d** Colocalization of IgG$^{Cy5.5}$ with R-M$^{FITC}$HAs in the absence or presence of NHS-TK-NHS. Scale bars: 10 μm. Data are represented as mean ± SD (*n* = 5 independent samples). *P* values were calculated using an unpaired two-tailed Student's *t* test. See also Supplementary Fig. 5 for complete data. **e** A schematic representation of cascade responsiveness of R-MAHAs to TME- and endosome-mimicking conditions.

The protein structure of ASNase was adapted from Cerofolini et al., with permission from John Wiley and Sons[58]. **f** Cumulative release curves of ASNase from R-MAHAs in PBS with or without 1 mM H$_2$O$_2$. **g** Changes in hydrodynamic diameters of R-MAHAs in PBS with or without 1 mM H$_2$O$_2$. **h, i** Cumulative release curves of Rot from R-MAHAs at pH 5.0, pH 6.7 and pH 7.4 with (**h**) or without 1 mM H$_2$O$_2$ (**i**). **f–i** Data are represented as mean ± SD (*n* = 3 independent samples). Asn asparagine, NPs nanoparticles, FITC fluorescein isothiocyanate, Cy5.5 cyanine 5.5, NHS-TK-NHS *N*-hydroxy succinimide-thioketal-*N*-hydroxy succinimide, TME tumor micro-environment, ASNase L-asparaginase, Rot rotenone. Source data are provided as a Source Data file.

observed in other solid tumor cell lines, including Panc02, LLC, and H22 tumor cells (Supplementary Fig. 12). Therefore, the R-MAHAs were able to sequentially consume extracellular Asn and inhibit de novo Asn synthesis to achieve more stringent Asn restriction, which could be utilized against tumor cells from solid tumors.

## Dual Asn-depriving NPs restrain tumor growth and metastasis in orthotopic TNBC

Next, we investigated the activities of R-MAHAs in a murine orthotopic 4T1 TNBC model, including tumor targeting and tumor suppression. We started with the in vivo pharmacokinetics analysis for Cy5.5-labeled M$^{Cy5.5}$AHAs. The half-life ($t_{1/2}$) of M$^{Cy5.5}$AHAs was 3.7 ± 0.7 h, which was over three times longer than that of free Cy5.5 (1.2 ± 0.2 h). This result indicated that drugs delivered by the MAHAs could significantly extend their blood circulation time (Supplementary Fig. 13). Compared to unmodified M$^{Cy5.5}$As, HA functionalization on the M$^{Cy5.5}$AHAs significantly enhanced NP accumulation in the tumors, as revealed by both in vivo and ex vivo fluorescence imaging. In the meanwhile, the major organs only displayed basal levels of NP accumulation (Fig. 4a and Supplementary Fig. 14). The confocal fluorescence imaging analysis confirmed that the M$^{Cy5.5}$AHAs were much more abundantly present on the tumor sections than the M$^{Cy5.5}$As, and injection of free Cy5.5 resulted in apparently lower levels of tumor retention than either type of NPs (Supplementary Fig. 15). Furthermore, we compared the destinies of M$^{Cy5.5}$As and M$^{Cy5.5}$AHAs in the tumor tissues. The M$^{Cy5.5}$As exhibited low levels of internalization by both tumor cells and tumor-infiltrating immune cells. In contrast, the M$^{Cy5.5}$AHAs showed a prominently high colocalization ratio with tumor cells rather than immune cells, verifying the specificity of HA molecules for tumor cell targeting in vivo (Fig. 4b). We continued to investigate the intratumor detachment of MAHAs. Dual-labeled M$^{FITC}$IgG$^{AF594}$HAs consisting of FITC-incorporated MHA cores (M$^{FITC}$HAs) and AlexaFluor 594 (AF594)-conjugated IgG shells (IgG$^{AF594}$) were fabricated to monitor core-shell dissociation. After tumor accumulation, M$^{FITC}$HAs and IgG$^{AF594}$ were well separated, as indicated by their low ratios of colocalization in the tumor section (Supplementary Fig. 16a). Flow cytometric analysis revealed a significant increase of M$^{FITC}$HA signals in the tumor cells after M$^{FITC}$IgG$^{AF594}$HA injection, whereas the intracellular IgG$^{AF594}$ signals remained low, further confirming the extracellular disintegration of protein shells (Supplementary Fig. 16b, c).

The 4T1 tumor-bearing mice were subjected to various treatments, including unloaded MHAs, free ASNase, free Rot, a combination of free ASNase and Rot (ASNase + Rot), and MAHAs loaded with an equal dose or double doses of Rot (R-MAHAs or R$^{2\times}$-MAHAs). These treatments were intravenously administered every 3 days and did not cause any fluctuation in the mouse body weight (Supplementary Fig. 17). Although free Rot was administered at a relatively low dose (0.4 mg/kg), it led to noticeable disturbance of serum liver function parameters such as alanine transaminase (ALT) and aspartate transaminase (AST), disclosing its systemic toxicity and thereby preventing our attempt on higher doses of free Rot. In contrast, no adverse effects on either liver or kidney function parameters were induced by the R$^{2\times}$-MAHAs, which were loaded with 0.8 mg/kg Rot and maintained the ASNase and NP mass unchanged (Supplementary Fig. 18). This data

revealed superior biocompatibility of MAHA-delivered Rot compared to its free form. Besides, R$^{2\times}$-MAHAs did not cause fluctuation of cardiac function parameters, as demonstrated by echocardiographic analysis of ejection fraction and fractional shortening, and serum levels of lactate dehydrogenase (LDH) and creatine kinase-MB (CK-MB) (Supplementary Fig. 19). We monitored dynamic tumor growth curves and evaluated the therapeutic effects of each treatment. By the time the mock-treated mice reached the end point of tumor growth, there was no restraint of tumor development by free ASNase or Rot, and a mild-to-moderate inhibitory effect was observed in the ASNase + Rot group. The R-MAHAs and R$^{2\times}$-MAHAs displayed obvious advantages over the free drug combination, reducing the mean tumor volume by 76.9 ± 3.6% and 87.9 ± 1.0%, respectively (Fig. 4c). Consistent with these results, combined drug delivery prolonged the mouse survival. The control mice displayed a median survival time of 24 days, which was hardly affected by free ASNase or Rot. Administration of ASNase + Rot, R-MAHAs and R$^{2\times}$-MAHAs incrementally extended the median survival time to 30, 48, and 60 days, respectively (Fig. 4d).

We explored the correlation between the therapeutic efficacy of above treatments and their Asn modulating capacities in the 4T1 tumors. Monotherapy with free ASNase or Rot failed to affect the intracellular Asn levels in isolated tumor cells, illustrating the necessity for dual Asn regulation against solid tumors. In consonance with their anti-tumor activities, the combination therapy with ASNase + Rot, R-MAHAs or R$^{2\times}$-MAHAs resulted in stepwise reduction of cytosolic Asn concentrations (Fig. 4e). Additionally, we examined the expression of ATF4 target genes (*ASNS, PSAT1, GPT2, MTHFD2,* and *SLC1A5*), which were activated upon sensing Asn insufficiency. As expected, treatment with R-MAHAs significantly upregulated the expression levels of aforementioned genes, and treatment with R$^{2\times}$-MAHAs showed more prominent effects (Fig. 4f and Supplementary Fig. 20). We further analyzed the TCA cycle metabolites after R$^{2\times}$-MAHA treatment. The cellular pools of fumarate, α-ketoglutarate (α-KG), aspartate, and the nicotinamide adenine dinucleotide (NAD$^+$)/NADH ratio were shrunken after R$^{2\times}$-MAHA treatment. In the meantime, R$^{2\times}$-MAHAs did not affect the cytosolic glutamate or methionine levels. Notably, similar effects were also observed in the R$^{2\times}$-MHA-treated group (Supplementary Fig. 21). We also measured extracellular Asn levels in the plasma and tumor interstitial fluid (TIF). Free Rot had no impact on Asn levels in either plasma or TIF. ASNase alone or in combination with Rot significantly reduced Asn levels in plasma and to a lesser extent in TIF. In contrast, treatment with R-MAHAs or R$^{2\times}$-MAHAs led to a moderate reduction of Asn levels in plasma and to a greater degree in TIF, indicating that drug delivery via the dual Asn-depriving NPs could greatly improve the efficacy of Asn-regulating drugs in the TME (Supplementary Fig. 22a, b). Besides, Asn levels in major organs (heart, liver, spleen, lung, kidney and brain) were not noticeably affected by R-MAHA or R$^{2\times}$-MAHA treatment. In comparison, ASNase or ASNase + Rot caused a mild decrease of Asn levels in heart, liver, spleen, and kidney (Supplementary Fig. 22c–h).

As Asn modulation in the TME may also remodel the immune microenvironment, we analyzed the immune cell populations in the tumors after treatment. Free Rot did not trigger CD8$^+$ T-cell recruitment, whereas ASNase or ASNase + Rot treatment led to a mild

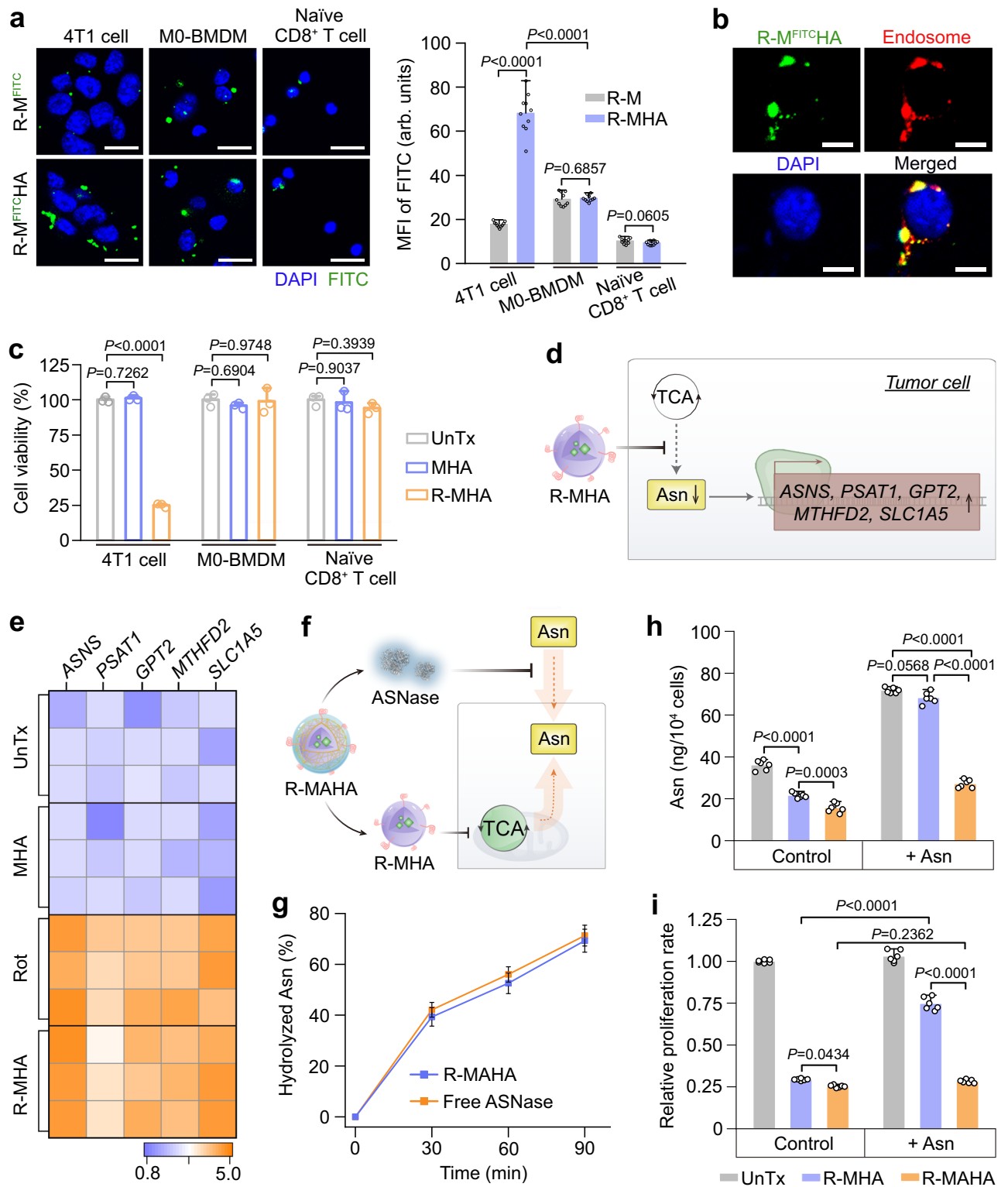

increase in the ratios of tumor-infiltrating CD8$^+$ T cells. R-MAHAs promoted CD8$^+$ T-cell infiltration with a superior efficiency to that of free drug combination (Supplementary Fig. 23a). Consistently, the R-MAHA treatment enhanced CD8$^+$ T-cell activation more potently than other treatments, as evidenced by elevated levels of tumor-infiltrating interferon-γ (IFN-γ)$^+$CD8$^+$ T cells and serum granzyme B (GZMB) (Supplementary Fig. 23b, c). The above effects were further augmented via R$^{2×}$-MAHAs. Similar results were found in the analysis of DC maturation makers, as indicated by the increased presenting of CD80 and CD86 molecules on the DC surfaces (Supplementary

Fig. 24). The activation of anti-tumor immunity following R-MAHA or R$^{2×}$-MAHA treatment was further confirmed by a significant decrease of regulatory T (Treg) cells and a mild reduction in the M2 type tumor associated macrophages (M2-TAMs) (Supplementary Fig. 25). Together, these results indicated that the dual Asn-depriving NPs could promote anti-tumor immune responses, which contributed to their therapeutic efficacy.

Histological examination of tissue sections from major organs (lung, heart, liver, kidney, and spleen) revealed that metastatic 4T1 tumors were frequently detected in the lungs of mock-treated mice. It

**Fig. 3 | Dual Asn modulation in vitro. a** Confocal fluorescence imaging (left) and quantifications (right) of R-micelles[FITC] (R-M[FITC]) and R-M[FITC]HAs in 4T1 cells, M0-BMDMs and naive CD8[+] T cells after 4 h of incubation. Scale bars: 20 μm. Data are represented as mean ± SD ($n = 10$ independent samples). $P$ values were calculated using an unpaired two-tailed Student's $t$ test. See also Supplementary Fig. 7 for complete data. **b** Intracellular localization of R-M[FITC]HAs. Scale bars: 10 μm. **c** Viability of 4T1 cells, M0-BMDMs and naive CD8[+] T cells after treatment with MHAs (15 μg/mL) or R-MHAs (MHAs: 15 μg/mL; Rot: 50 ng/mL) for 48 h. Data are represented as mean ± SD ($n = 3$ independent samples). $P$ values were calculated using a one-way ANOVA followed by Tukey's post-hoc test. See also Supplementary Fig. 10 for complete data. **d** Schematic illustration of upregulated transcription of *ASNS*, *PSAT1*, *GPT2*, *MTHFD2*, and *SLC1A5* in response to R-MHA treatment. **e** Heatmap of mRNA levels of *ASNS*, *PSAT1*, *GPT2*, *MTHFD2*, and *SLC1A5* in 4T1 cells after the indicated treatment. **f** Schematic illustration depicting the restriction of both intracellular and extracellular Asn to inhibit tumor cell proliferation. **g** Efficiency of Asn hydrolysis by ASNase (0.6 μg/mL) released from $H_2O_2$-stimulated R-MAHAs. Data are represented as mean ± SD ($n = 4$ independent samples). **h**, **i** Intracellular levels of Asn (**h**) and relative proliferation rate of 4T1 cells (**i**) after the indicated treatment, with or without supplementing Asn (0.1 mM) in the cell culture medium. R-MHAs (Rot: 50 ng/mL) and R-MAHAs (Rot: 50 ng/mL; ASNase: 0.6 μg/mL) were used. Data are represented as mean ± SD ($n = 6$ independent samples). $P$ values were calculated using a one-way ANOVA followed by Tukey's post-hoc test. Asn asparagine, FITC fluorescein isothiocyanate, BMDM bone marrow-derived macrophages, TCA tricarboxylic acid, ASNase L-asparaginase. Source data are provided as a Source Data file.

was reported that tumor metastasis was more sensitive to Asn restriction than tumor growth[19]. Indeed, administration of free ASNase or Rot could reduce the number of metastatic tumor foci by half, and their combination further enhanced this effect. Encouragingly, treatment with either R-MAHAs or R[2×]-MAHAs completely abrogated lung metastasis of the 4T1 tumors (Fig. 4g and Supplementary Figs. 26 and 27). Epithelial–mesenchymal transition (EMT) plays a decisive role in tumor metastasis. This process is accompanied with upregulation in the levels of key transcription factor Snail and mesenchymal cell marker N-cadherin, and downregulation in the levels of epithelial cell marker E-cadherin[50–53]. Immunohistochemistry (IHC) staining of tumor sections revealed that the combination of Rot and ASNase dramatically downregulated N-cadherin and Snail levels and largely boosted E-cadherin levels in the primary tumors, with the R[2×]-MAHAs showing the highest efficiency (Fig. 4h, i and Supplementary Fig. 28). These data suggest that the dual Asn-depriving NPs can remarkably reverse the process of EMT in solid tumors, which holds a strong potential in the prevention of tumor metastasis.

### Suppression of recurrence, metastasis and tumor rechallenge in post-surgical TNBC

Surgical removal of primary tumors may accelerate and aggravate metastasis[54–57]. Inspired by the effectiveness of dual Asn restriction in suppressing the growth and spontaneous metastasis of primary tumors, we continued to assess its efficacy in the post-surgical relapse and metastasis of TNBC. Various treatments were applied every 3 days after tumor resection (Fig. 5a). Real-time tracking of tumor progression by in vivo bioluminescence imaging displayed that the mock-treated mice quickly developed local recurrence and distant metastasis. The administration of ASNase + Rot mildly delayed post-surgical tumor progression. In comparison, R-MAHAs and R[2×]-MAHAs exhibited much stronger suppressive effects, preventing tumor relapse in one-third and two-third of the treated mice, respectively. The rest of the mice receiving the NP treatments showed obvious postponement in the onset of tumor recurrence with much slower growth rates. In addition, no metastatic signal was observed in the mice after treatment with R-MAHAs or R[2×]-MAHAs (Fig. 5b, c). Ex vivo bioluminescence analysis of major organs further disclosed that metastatic foci were frequently formed in the lung and to a lesser extent in the liver, kidney and heart in the control mice. Administration of R-MAHAs or R[2×]-MAHAs totally eliminated the metastatic signals in these organs, which was also confirmed by hematoxylin-eosin (H&E) staining of the tissue sections (Fig. 5d and Supplementary Fig. 29).

We measured the alterations of Asn metabolism in the relapsed tumors. Treatment with ASNase + Rot, R-MAHAs or R[2×]-MAHAs resulted in degressive Asn levels in the tumor cells (Supplementary Fig. 30). In response to the decrease of intracellular Asn reservoir, the expression of ATF4-driven metabolic genes was markedly elevated by R-MAHAs and to a greater degree by R[2×]-MAHAs (Supplementary Fig. 31). The improvement in the efficiency of Asn restriction by the NPs also led to significant extension of mouse survival. The median survival

time for the control group and ASNase + Rot group was 36 and 50 days, respectively. While the administration of R-MAHAs extended the median survival time to 72 days, the R[2×]-MAHA treatment showed the most beneficial effect with four out of six mice in this group surviving over 86 days (Fig. 5e). Besides, neither R[2×]-MAHAs nor R-MAHAs caused any adverse effect on the mouse body weight (Supplementary Fig. 32). Together, these data supported superior efficacies of dual Asn-depriving NPs in the suppression of post-surgical tumor recurrence and metastasis in a TNBC model.

We further explored the long-term protective effects of the dual Asn-depriving therapy and performed a tumor rechallenge study in the mice that had been healed by the post-surgical R[2×]-MAHA treatment. In vivo bioluminescence imaging disclosed that all the age-matched naive mice underwent fast tumor progression following tumor cell inoculation, whereas all the cured mice exhibited complete resistance to the tumor cell re-injection and no bioluminescence signal was detected 1 month after the rechallenge (Fig. 5f). Subsequently, we analyzed central memory T (Tcm) cells in the spleen that could confer the anti-tumor immune memory response. We observed a strong induction in the levels of both CD4[+] and CD8[+] Tcm cells as well as the serum levels of IFN-γ and IL-2 in the cured mice compared to the naive mice (Fig. 5g, h and Supplementary Fig. 33). Therefore, in addition to inhibiting tumor growth and preventing tumor recurrence and metastasis, the dual Asn-depriving NPs built and maintained long-term T-cell memory, which could help the host generate fast recall responses and reject tumor rechallenge.

### Applicability to post-surgical CRC

We evaluated the general applicability of the dual Asn-depriving NPs in solid tumors using a post-surgical CT26 CRC model (Fig. 6a). The modulation of Asn metabolism by various treatments was analyzed in the relapsed CT26 tumors. ASNase + Rot, R-MAHAs and R[2×]-MAHAs demonstrated escalating capacities in depleting the intracellular Asn reservoirs (Fig. 6b). In agreement with this observation, the R[2×]-MAHAs activated the ATF-controlled metabolic genes more efficiently than the R-MAHAs, which was more efficient than the free drug combination (Fig. 6c and Supplementary Fig. 34). The alterations in Asn metabolism were closely correlated with the inhibition of EMT. Immunofluorescence analysis of the tumor sections suggested that the R-MAHA treatment evidently upregulated the levels of epithelial cell marker E-cadherin and noticeably diminished the levels of mesenchymal cell marker N-cadherin, and both effects were further strengthened by the R[2×]-MAHA treatment (Fig. 6d, e and Supplementary Fig. 35).

Next, we examined the efficiency of dual Asn-depriving therapy in the prevention of post-surgical tumor relapse and metastasis. All the mice receiving the mock treatment experienced tumor recurrence and rapid progression shortly after surgery, as illustrated by in vivo bioluminescence imaging and tumor growth curves. The free drug combination failed to prevent tumor relapse, but slightly mitigated tumor development. The R-MAHAs prevented post-surgical recurrence in two

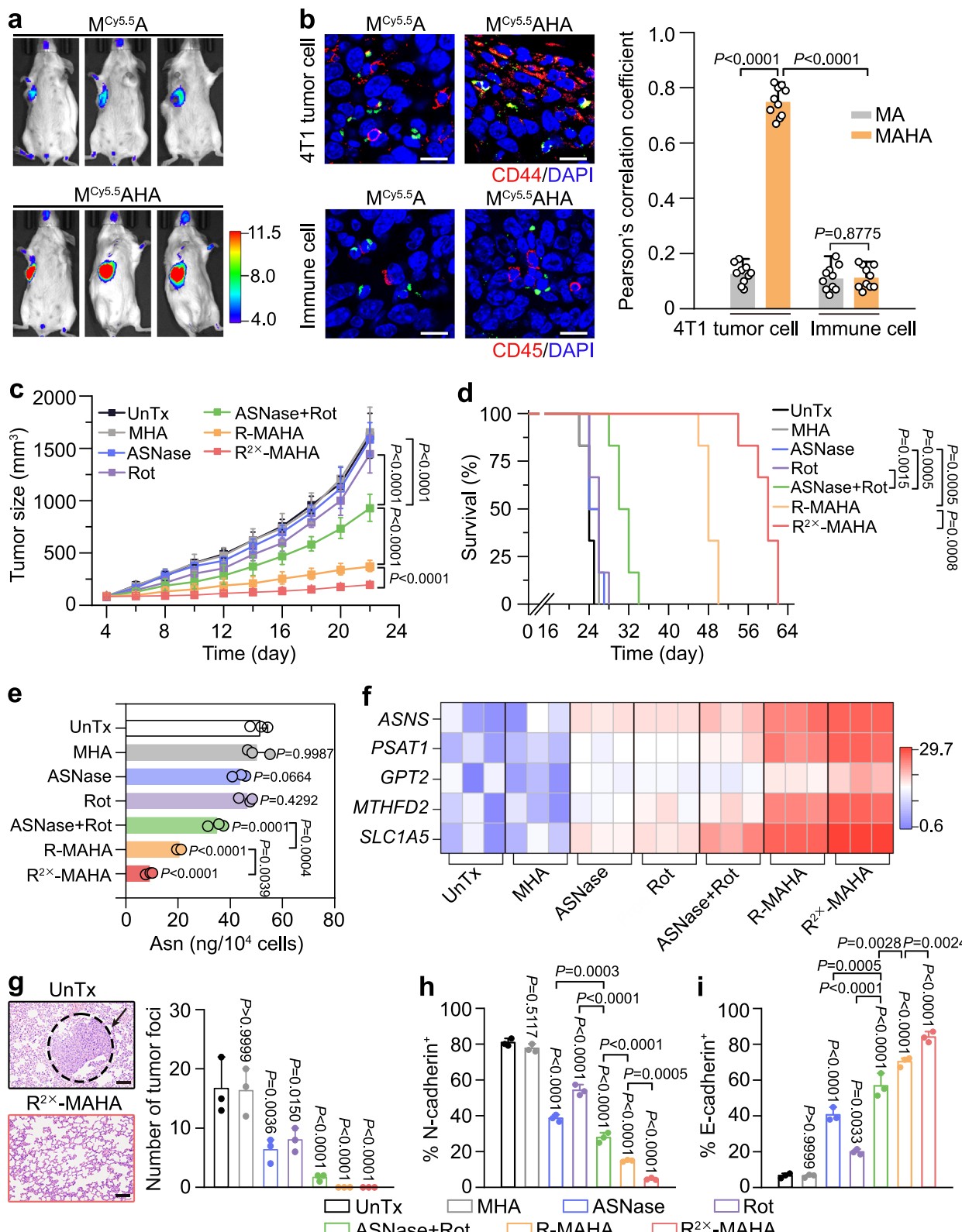

out of six mice and prominently delayed tumor progression in the rest of mice. This efficacy was significantly enhanced by the R²ˣ-MAHA treatment, avoiding tumor recurrence in five out of six mice (Fig. 6f and Supplementary Fig. 36). We monitored tumor metastasis in the intestine and other major organs via ex vivo bioluminescence imaging. The mice administered with PBS, unloaded MHAs, or the free drug combination showed frequent metastasis in the intestine and to a lesser degree in the liver, spleen, and lung. In contrast, metastatic signals were thoroughly absent in the organs of the mice treated with R-MAHAs or R²ˣ-MAHAs (Fig. 6g and Supplementary Fig. 37). Similarly, the control mice exhibited extensive intestinal metastasis and sporadic liver and lung metastasis in the histological examination of tissue sections, which were eradicated in the mice treated with R-MAHAs or R²ˣ-MAHAs (Supplementary Fig. 38).

**Fig. 4 | Efficacy of dual Asn-depriving NPs in orthotopic TNBC. a** In vivo fluorescence imaging of tumor-bearing mice 24 h after intravenous injection of $M^{Cy5.5}As$ and $M^{Cy5.5}AHAs$ ($n = 3$ mice). The unit of the scale bar is ×10^9 [p/s/cm²/sr]/[μW/cm²]. **b** Left: Confocal fluorescence imaging of tumor sections after administration of $M^{Cy5.5}As$ and $M^{Cy5.5}AHAs$. The 4T1 tumor cells and immune cells were indicated as CD44 and CD45 positive cells (red fluorescence), respectively. $M^{Cy5.5}As$ and $M^{Cy5.5}AHAs$ were pseudo-colored with green fluorescence. DAPI (blue fluorescence) was used to label the nuclei. Scale bars: 15 μm. Right: Values of Pearson's correlation coefficient between Cy5.5-incorporated NPs and 4T1 tumor cells or immune cells. Data are represented as mean ± SD ($n = 10$ independent samples). *P* values were calculated using an unpaired two-tailed Student's *t* test. **c** Average tumor growth curves. Data are represented as mean ± SD ($n = 6$ mice). *P* values were calculated using a two-way ANOVA followed by Tukey's multiple comparisons test. **d** Survival curves of the mice receiving indicated treatment ($n = 6$ mice). *P* values were calculated using a log-rank (Mantel−Cox) test. **e** The Asn levels in tumor cells isolated from tumors on day 26 after indicated treatment. Data are represented as mean ± SD ($n = 3$ independent samples). *P* values were calculated using a one-way ANOVA followed by Tukey's post-hoc test. **f** Heatmap of mRNA levels of *ASNS*, *PSAT1*, *GPT2*, *MTHFD2*, and *SLC1A5* in tumor cells isolated from tumors after indicated treatment. **g** Left: representative H&E-stained sections of lungs from indicated groups. Metastatic tumors are indicated with black dotted circles. Scale bars: 100 μm. See also Supplementary Fig. 26 for complete data. Right: the number of tumor foci in lungs after indicated treatment. Data are represented as mean ± SD ($n = 3$ mice). *P* values were calculated using a one-way ANOVA followed by Tukey's post-hoc test. **h, i** Expression levels of N-cadherin (**h**) and E-cadherin (**i**) in tumors after indicated treatments. Data are represented as mean ± SD ($n = 3$ independent samples). *P* values were calculated using a one-way ANOVA followed by Tukey's post-hoc test. Asn asparagine, Cy5.5 cyanine 5.5, Rot rotenone, ASNase L-asparaginase. Source data are provided as a Source Data file.

We further measured the survival time of the mice receiving above treatments. The median survival time for the mice receiving mock treatment, unloaded MHA treatment and ASNase + Rot treatment was 46, 44, and 54 days, respectively. The R-MAHA treatment extended the median survival time to 66 days. Notably, five out of six mice receiving the R²ˣ-MAHA treatment stayed alive at the end of our observation (Fig. 6h). None of the treatments caused fluctuation in the body weight (Supplementary Fig. 39). Lastly, we employed tumor rechallenge to test T-cell memory in the recovered mice and all of them exhibited total rejection to inoculated CT26 CRC cells in a 1-month observation window, suggesting the establishment of long-term protective immunity in these mice (Supplementary Fig. 40a). Consistently, subsequent analysis of Tcm cells in the spleen disclosed that the cured mice possessed much higher levels of CD4⁺ and CD8⁺ Tcm cells compared to the age-paired control mice (Supplementary Fig. 40b, c). Moreover, both CD4⁺ and CD8⁺ Tcm cells exhibited higher levels of Ki-67 expression in cured mice compared to the naive mice, indicating enhanced proliferative capacity (Supplementary Fig. 41a, b). In addition, tumor rechallenge in cured mice strongly stimulated Tcm cell activation, as evidenced by increased IFN-γ and IL-2 secretion in the serum (Supplementary Fig. 41c, d). Accordingly, these results demonstrated great therapeutic efficiency of dual Asn-depriving NPs against post-surgical relapse, metastasis, and tumor rechallenge in a CRC model.

## Discussion

The dual Asn-depriving NPs possess several advantages in the treatment of solid tumors. First, the endogenous Asn synthetic pathway in solid tumors can provide sufficient Asn reservoir to satisfy the requirement for tumor growth, and thereby renders the ineffectiveness of extracellular-acting ASNase monotherapy. The R-MAHAs and R²ˣ-MAHAs are designed to release ASNase in the TME for the hydrolysis of extracellular Asn and Rot inside tumor cells for the blockade of Asn biosynthesis, cutting off both exogenous and endogenous sources of Asn supply. Consequently, treatment with the dual-Asn-depriving NPs, especially R²ˣ-MAHAs, resulted in substantial inhibition of tumor growth and post-surgical recurrence. Second, the intracellular levels of Asn are closely correlated with EMT and tumor metastasis due to the high Asn content in the EMT-promoting proteins[19,20]. In comparison to free ASNase, free Rot or their combination, both R-MAHAs and R²ˣ-MAHAs displayed remarkable capacities in reversing the levels of EMT marker proteins (i.e., Snail, N-cadherin, and E-cadherin) in primary and relapsed tumors, and completely eliminated spontaneous and post-surgical metastasis. Third, the effects of systemic Asn deprivation on T cells are controversial[21–23]. The tumor-targeting NPs can minimize the off-target effects of Asn modulation in T-cell generating, processing and storage organs and tissues. As a result, administration of R-MAHAs or R²ˣ-MAHAs stimulated the anti-tumor responses of tumor-infiltrating CD8⁺ T cells. Moreover, the dual Asn-depriving NPs have superior biocompatibility. We observed the disturbance of liver function parameters by free Rot, although it was administered at a relatively low dose, restricting its potential in clinical application. By contrast, the R-MAHAs or R²ˣ-MAHAs loaded with equal or double doses of Rot did not induce any adverse effect while accomplishing much better therapeutic outcomes.

In conclusion, we have designed a type of cascade-responsive NPs with dual capacities for sequential modulation of external Asn supply and internal Asn generation, thereby achieving stringent Asn depletion in solid tumors. Systemic administration of these NPs in murine models of TNBC and CRC showed superior efficacy in inhibiting tumor growth and relapse, preventing EMT and metastasis, and conferring long-term protective immunity against tumor rechallenge. These findings provide a highly promising strategy to develop broadly applicable cancer metabolic therapy based on amino acid modulation.

## Methods

### Ethics statement

All mouse experiments were reviewed and approved by the Animal Care and Use Committee of Shanghai Jiao Tong University School of Medicine (A-2022-115). A maximum tumor size of 20 mm in any dimension, as approved by the ethics committee, was not exceeded during the study. The mice were euthanized when their tumors reached a volume of ~2000 mm³, with any dimension below 20 mm.

### Preparation and characterization of NPs

PCL-Hyd-PEG-NH₂ (Ruixibio) was conjugated with HA using 1-(3-dimethylaminopropyl)-3-ethylcarbodiimide hydrochloride (EDC) and N-hydroxysuccinimide (NHS) (Sigma). Specifically, 1 mg HA (MedChemExpress) was activated with 0.5 mg NHS and 5 mg EDC in dimethyl sulfoxide, followed by incubation with 1 mg PCL-Hyd-PEG-NH₂ for 6 h. To prepare micellar cores, 9 mg undecorated PCL-Hyd-PEG-NH₂ was dissolved in the above mixture, and then 10 mL ultrapure water was added dropwise. After stirring overnight at room temperature, the self-assembled MHAs were collected by centrifugation at 10,000×*g* for 15 min and washed twice with ultrapure water to remove excess EDC/NHS and unconjugated HA. For the preparation of Rot-loaded MHAs, Rot (MedChemExpress) was concurrently dissolved in the mixture. The encapsulated Rot in R-MHAs was quantified by UV–Vis-NIR absorbance spectra (UV-1800, SHIMADZU).

For the construction of ASNase shells, R-MHAs were incubated with 0.67 mg NHS-TK-NHS (Ruixibio) for 30 min at room temperature. Unconjugated NHS-TK-NHS molecules were removed by centrifugation and washing steps. The precipitates were then resuspended in 1 mL PBS, and 0.67 mg ASNase (MedChemExpress) dissolved in 1 mL PBS was added dropwise. The reaction mixture was rotated at room temperature for an additional 30 min. Finally, the resulting R-MAHAs were collected by centrifugation and washed twice with PBS to remove unbound ASNase. The hydrodynamic diameter and zeta potential of

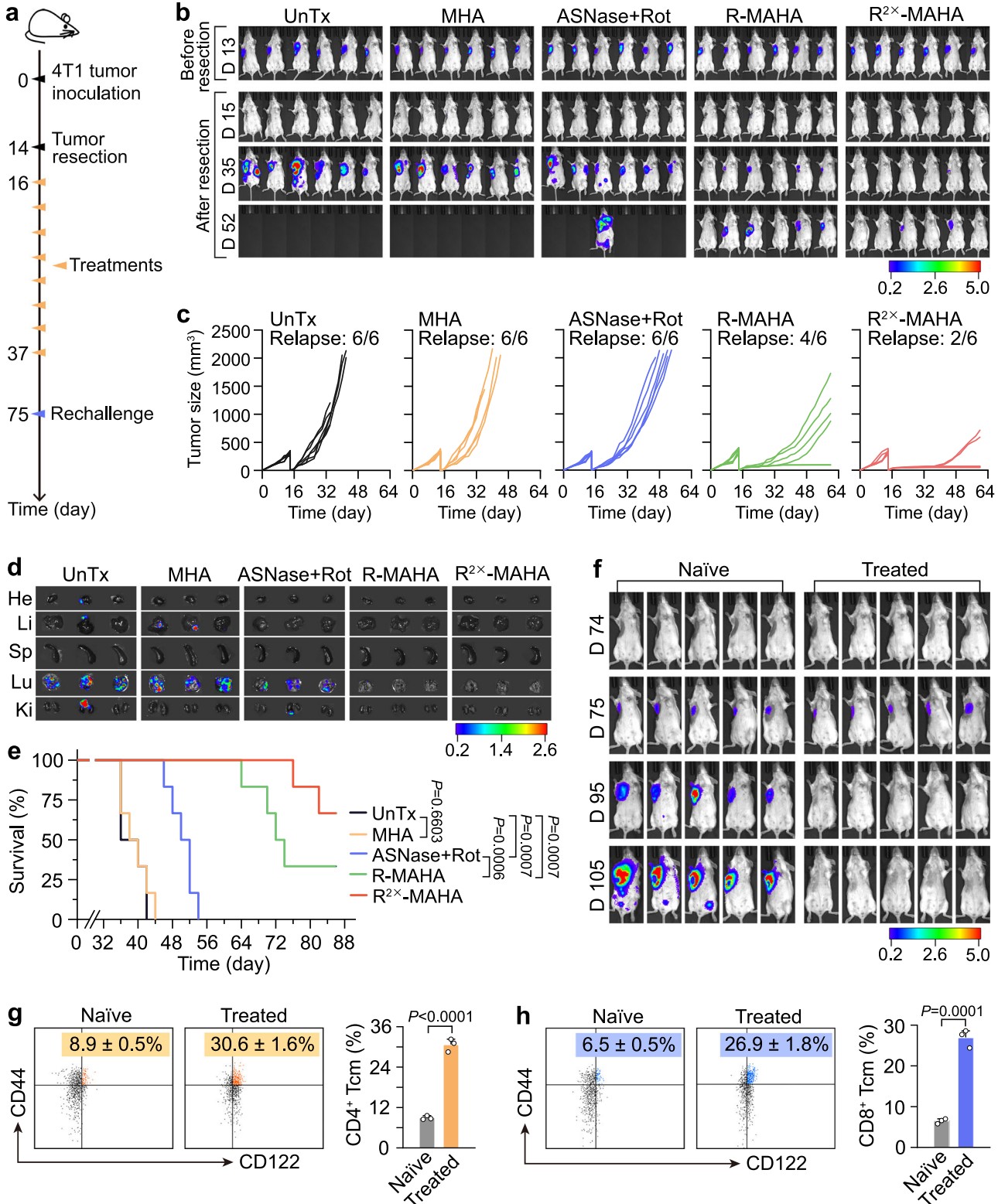

**Fig. 5 | Prevention of relapse and metastasis in post-surgical TNBC model.**
**a** Schematic illustration of the experiment design for post-surgical therapy. Fourteen days after tumor inoculation, primary tumors were removed. Two days later, mice were subjected to various treatments every 3 days for a total of eight doses. On day 75, mice with no sign of tumor relapse and metastasis were challenged with 4T1-Luc cells. **b** In vivo bioluminescence imaging of the mice after indicated treatment ($n = 6$ mice). The unit of the scale bar is ×10⁷ p/s/cm²/sr. **c** Tumor growth curves of individual mice after indicated treatment ($n = 6$ mice). **d** Ex vivo bioluminescence images of major organs ($n = 3$ mice). The unit of the scale bar is ×10⁶ p/s/cm²/sr. **e** Survival curves of the mice receiving the indicated treatment ($n = 6$

mice). P values were calculated using a log-rank (Mantel–Cox) test. **f** In vivo bioluminescence imaging of mice before (day 74) and after (day 75, day 95, and day 105) tumor rechallenge ($n = 5$ mice). The unit of the scale bar is ×10⁷ p/s/cm²/sr. **g**, **h** FACS analysis of CD4⁺ Tcm (gated on CD45⁺CD4⁺ population) (**g**) and CD8⁺ Tcm cells (gated on CD45⁺CD8⁺ population) (**h**) in splenocytes. Data are represented as mean ± SD ($n = 3$ independent samples). P values were calculated using an unpaired two-tailed Student's t test. Rot rotenone, ASNase L-asparaginase, He heart, Li liver, Sp spleen, Lu lung, Ki kidney, Tcm central memory T. Source data are provided as a Source Data file.

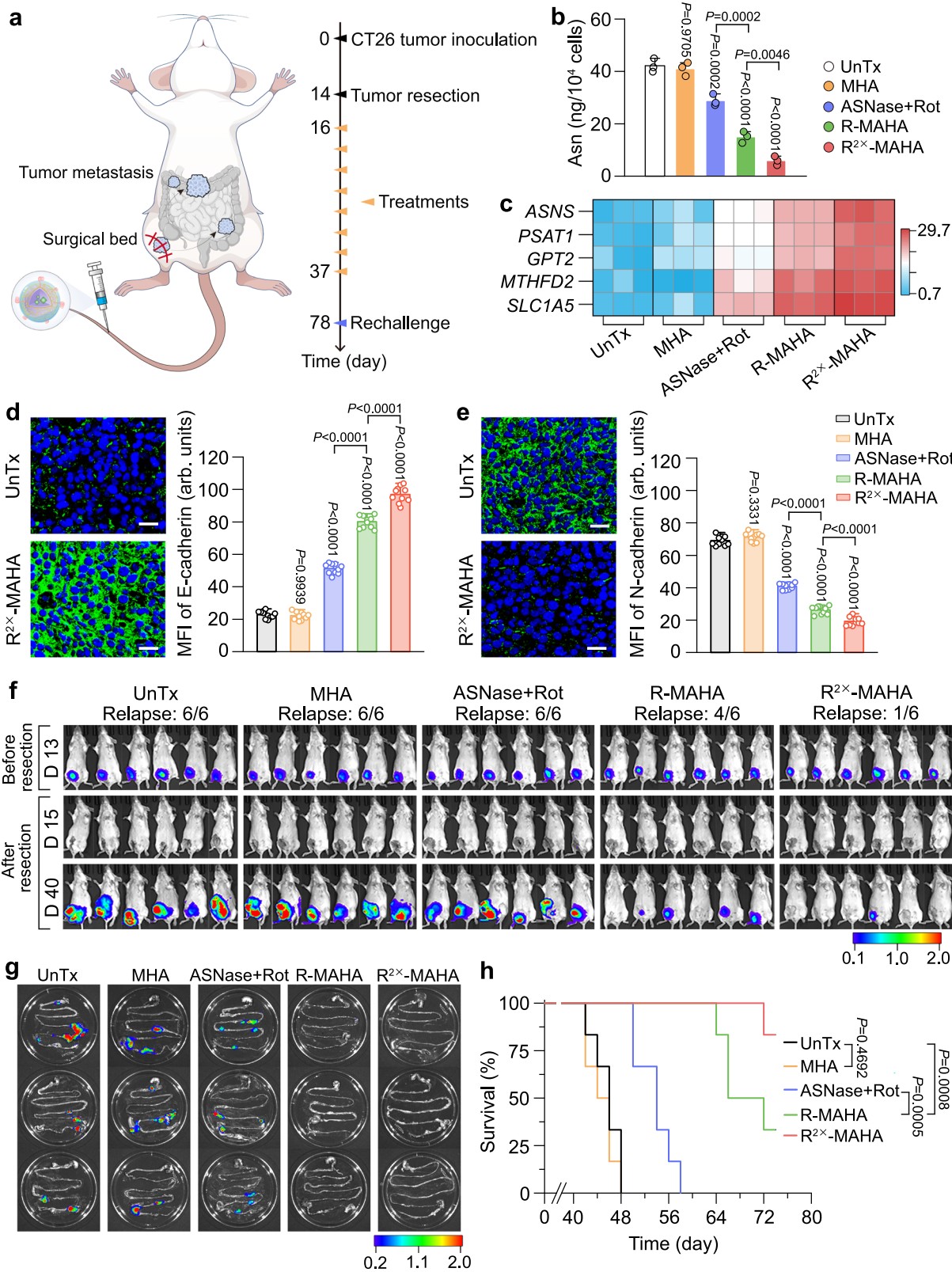

NPs were characterized by ZEN3690 Zetasizer (Malvern), and their morphology was characterized by TEM (JEM-1200EX, JEOL, Ltd., Japan).

**Releasing kinetics of ASNase and Rot**
To study the disintegration of ASNase shells, R-MAHAs were dispersed in PBS buffer with or without 1 mM $H_2O_2$. Supernatants were collected at various time points, and the amount of ASNase released was quantified using a bicinchoninic acid (BCA) protein assay kit. To evaluate pH-responsive Rot release from the micellar cores, R-MAHAs were first treated with 1 mM $H_2O_2$ for 2 h, followed by incubation in PBS buffer at different pH levels (5.0, 6.7, and 7.4). The amount of released Rot was measured using UV-vis-NIR absorbance spectroscopy.

**Fig. 6 | Applicability of dual Asn deprivation in post-surgical CRC model.**
**a** Schematic illustration of the therapeutic schedule. Fourteen days after tumor inoculation, the primary tumors were removed. Two days later, the mice were subjected to various treatments every 3 days for a total of eight doses. On day 78, the mice with no sign of tumor relapse and metastasis were challenged with CT26-Luc cells. Created in BioRender. shen, y. (2025) https://BioRender.com/f8okx09. **b** The Asn levels in tumor cells isolated from relapsed CRC on day 38 after indicated treatment. Data are represented as mean ± SD ($n = 3$ independent samples). $P$ values were calculated using a one-way ANOVA followed by Tukey's post-hoc test. **c** Heatmap of mRNA levels of *ASNS*, *PSAT1*, *GPT2*, *MTHFD2*, and *SLC1A5* in tumor cells isolated from relapsed CRC after indicated treatment. **d**, **e** Representative

immunofluorescence images and quantifications of E-cadherin (**d**) and N-cadherin (**e**) (green fluorescence) in the tumor sections after indicated treatment. DAPI (blue fluorescence) was used to label the nuclei. Scale bars: 20 μm. Data are represented as mean ± SD ($n = 10$ independent samples). $P$ values were calculated using a one-way ANOVA followed by Tukey's post-hoc test. See also Supplementary Fig. 35 for complete data. **f** In vivo bioluminescence imaging of the mice after indicated treatment ($n = 6$ mice). The unit of the scale bar is ×$10^7$ p/s/cm²/sr. **g** Ex vivo bioluminescence images of intestines after indicated treatment ($n = 3$ mice). The unit of the scale bar is ×$10^7$ p/s/cm²/sr. **h** Survival curves of the mice receiving indicated treatment ($n = 6$ mice). $P$ values were calculated using a log-rank (Mantel–Cox) test. Asn asparagine, Rot rotenone, ASNase L-asparaginase. Source data are provided as a Source Data file.

## Cell culture

4T1 breast cancer cells (Serial: TCM32), CT26 colorectal cancer cells (Serial: TCM37), and LLC lung cancer cells (Serial: TCM47) were obtained from the Cell Bank of the Chinese Academy of Sciences (Shanghai). Panc02 pancreatic cancer cells (Serial: CVCL_D627) and H22 hepatic cancer cells (Serial: CVCL_H613) were obtained from Fuheng Biology Science and Technology Co., Ltd. (Shanghai). All cell lines were authenticated by short tandem repeat (STR) analysis and tested for mycoplasma routinely. LLC and Panc02 cells were cultured in Dulbecco's modified Eagle medium (DMEM, Gibco). 4T1, CT26, and H22 cells were cultured in RPMI-1640 medium (Gibco). All media contained 10% fetal bovine serum (FBS, Gibco) and 100 U/mL streptomycin/penicillin (Invitrogen). To prepare CD8+ T cells and BMDMs, female BALB/c mice aged at 6–8 weeks were purchased from SLAC Laboratory Animal Co. Ltd (Shanghai, China). Naive CD8+ T cells were isolated from BALB/c mice using the MojoSort™ Mouse CD8+ T Cell Isolation Kit (BioLegend) and cultured in the RPMI-1640 medium supplemented with 10% FBS and 100 U/mL streptomycin/penicillin. To induce activated CD8+ T cells, naive CD8+ T cells were stimulated with anti-CD3ε (2 μg/mL, BioLegend) and anti-CD28 (2 μg/mL, BioLegend) for 48 h. To generate M0-BMDMs, bone marrow cells were isolated from BALB/c mice and stimulated with 20 ng/mL granulocyte–macrophage colony-stimulating factor (GM-CSF, eBioscience) for 7 days, and the medium was refreshed on day 3.

## Targeting capability in vitro

FITC-labeled micelles (R-micelles^FITC and R-M^FITCHAs) were prepared to study the targeting capacity of HA. 4T1 cells, M0-BMDMs, naive CD8+ T cells and activated CD8+ T cells were incubated with R-micelles^FITC or R-M^FITCHAs for varying durations (1, 2, and 4 h). Intracellular fluorescence was visualized using confocal laser scanning microscopy (CLSM, Leica TCS SP8), with consistent instrument settings applied for all images.

To monitor the intracellular localization of R-M^FITCHAs, 4T1 cells were incubated with R-M^FITCHAs for 6 h, followed by staining with LysoTracker™ Deep Red (250 nM, ThermoFisher Scientific, L12492) for 40 min. The cells were then fixed with 4% paraformaldehyde (PFA), washed twice with PBS, and stained with 4′,6-diamidino-2-phenylindole (DAPI, 2.5 μg/mL).

## Cell viability assay

4T1 cells were cultured in medium with or without 0.1 mM Asn and treated with R-MHAs (MHA: 15 μg/mL; Rot: 50 ng/mL) or R-MAHAs (MHAs: 15 μg/mL; Rot: 50 ng/mL; ASNase: 0.6 μg/mL) for 48 h. To facilitate ASNase release, R-MAHAs were pre-treated with 1 mM $H_2O_2$. Cell viability was assessed by the Cell Counting Kit-8 (CCK-8, Beyotime).

## Animal tumor models and treatment

Female BALB/c mice aged at 6–8 weeks were purchased from SLAC Laboratory Animal Co. Ltd (Shanghai, China). The mice were kept in a barrier environment with a constant temperature of 24 °C and a relative humidity of 50%. The mice were maintained under a 12-h light and 12-h dark cycle.

To establish an orthotopic TNBC model, $2 \times 10^5$ 4T1 cells suspended in 100 μL of Matrigel (Corning) were injected into the mammary fat pads of mice. Once the tumors reached ~80 mm³, mice received intravenous administration of PBS, unloaded MHAs (35 mg/kg), free ASNase (400 IU/kg), free Rot (0.4 mg/kg), ASNase + Rot (ASNase: 400 IU/kg; Rot: 0.4 mg/kg), R-MAHAs (MHAs: 35 mg/kg; ASNase: 400 IU/kg; Rot: 0.4 mg/kg), or R²ˣ-MAHAs (MHAs: 35 mg/kg; ASNase: 400 IU/kg; Rot: 0.8 mg/kg) every 3 days for a total of eight doses.

To establish the post-surgical TNBC model, $3 \times 10^5$ 4T1-Luc cells (expressing firefly luciferase via recombinant lentiviral infection) were suspended in 100 μL of Matrigel and injected into the mammary fat pads of mice. Once the tumors reached ~400 mm³, primary tumors were surgically resected. Two days after surgery, the mice were administered intravenously with PBS, unloaded MHAs (35 mg/kg), free ASNase (400 IU/kg), free Rot (0.4 mg/kg), ASNase + Rot (ASNase: 400 IU/kg; Rot: 0.4 mg/kg), R-MAHAs (MHAs: 35 mg/kg; ASNase: 400 IU/kg; Rot: 0.4 mg/kg), or R²ˣ-MAHAs (MHAs: 35 mg/kg; ASNase: 400 IU/kg; Rot: 0.8 mg/kg) every 3 days, for a total of eight doses. For the post-surgical CRC model, $1 \times 10^6$ CT26-Luc cells were suspended in 100 μL of Matrigel and injected into the left groin of mice. The mice then received similar treatments as described above. For the tumor rechallenge experiment, $3 \times 10^5$ 4T1-Luc cells or $1 \times 10^6$ CT26-Luc cells were injected into the mammary fat pads or left groin of cured mice, respectively. Age-matched mice served as the control group.

Tumor progression was monitored using in vivo bioluminescence imaging with the IVIS Spectrum Imaging System (PerkinElmer) following intraperitoneal injection of D-Luciferin (150 μg/g). The tumor size was measured with calipers, and tumor volume was calculated as follows: tumor volume = $0.5 \times$ length $\times$ width².

## In vivo pharmacokinetics

When the tumor volumes reached ~500 mm³, mice were intravenously injected with either free Cy5.5 or M^Cy5.5AHAs at a dose of 1 mg/kg Cy5.5. Blood samples were collected from the severed tails at designated time points and treated with 1% heparin. Cy5.5 concentrations in the samples were then measured using a microplate reader.

## In vivo and ex vivo fluorescence imaging

When the tumor volumes reached approximately 500 mm³, the mice received intravenous injections of either M^Cy5.5As or M^Cy5.5AHAs (Cy5.5: 1 mg/kg) once a day for a total of three doses. In vivo imaging was conducted using a PerkinElmer imaging system 24 h after the final injection. Next, the tumors and major organs were excised for ex vivo fluorescence imaging. The tumors were then used to prepare frozen sections. Tumor sections were stained with either anti-CD44 rabbit polyclonal antibody (Servicebio, GB112054) or anti-CD45 rabbit polyclonal antibody (Servicebio, GB113885), and visualized using CLSM. To monitor core-shell dissociation in vivo, dual fluorescence-labeled M^FITCIgG^AF594HAs (IgG^AF594: AlexaFluor™ 594 goat anti-rabbit secondary antibody (Invitrogen, A11012)) were intravenously injected in tumor-bearing mice once a day for a total of three doses. The tumors were harvested for the preparation of frozen sections and imaged under CLSM.

## Analysis of amino acids and TCA cycle metabolites

The 4T1 cells were cultured in medium with or without 0.1 mM Asn supplementation and treated with R-MHAs (MHAs: 15 µg/mL; Rot: 50 ng/mL) or R-MAHAs (MHAs: 15 µg/mL; Rot: 50 ng/mL; ASNase: 0.6 µg/mL) for 48 h. The R-MAHAs were pre-treated with 1 mM $H_2O_2$ to facilitate ASNase release. Intracellular Asn was extracted with cell lysis buffer and homogenizer (SCIENTZ-II D). The Asn levels in the supernatants were measured with an Asn content assay kit (Geruisi-bio, G0437W). For in vivo analysis, the 4T1 and CT26 tumors were digested into single-cell suspensions by the Tumor Dissociation Kit (Miltenyi) and sorted by flow cytometry. The tumor cells were then lysed for the analysis of intracellular Asn levels and TCA cycle metabolites (fumarate: Beyotime, S0517S; α-KG: Solarbio, BC5425; NAD$^+$/NADH: Beyotime, S0175). Cytosolic levels of aspartate, glutamate, and methionine were assessed by High-Speed Amino Acid Analyzer (Hitachi, LA8080 AminoSAAYA).

## Real-time PCR analysis

Total RNAs were extracted from tumor cells using the TRIzol reagent (Invitrogen) to analyze the expression levels of *ASNS*, *PSAT1*, *GPT2*, *MTHFD2*, and *SLC1A5*. *GAPDH* served as the internal control to normalize the relative expression levels of the target genes.

## Immune cell analysis

For analysis of CD8$^+$ T cells, DCs, Treg cells, and M2-TAMs, the 4T1 tumors were collected from the mice on day 26 and digested into single-cell suspensions. The following antibodies were used: APC/Cy7-conjugated anti-CD45 (BioLegend, 103116), PerCP/Cy5.5-conjugated anti-CD8 (BD, 551162), PE-conjugated anti-CD4 (BioLegend, 100408), PE/Cy7-conjugated anti-IFN-γ (BioLegend, 505826), PE-conjugated anti-CD11c (BioLegend, 117308), PerCP/Cy5.5-conjugated anti-CD80 (BioLegend, 104721), BV421-conjugated anti-CD86 (BioLegend, 105031), BV421-conjugated anti-CD25 (BD Horizon, 562606), PE/Cy7-conjugated anti-Foxp3 (eBioscience, 25577382), PerCP/Cy5.5-conjugated anti-CD11b (BioLegend, 101228), PE-conjugated anti-F4/80 (BioLegend, 123110), and BV421-conjugated anti-CD206 (BioLegend, 141717). The CD8$^+$ T cells, IFN-γ$^+$CD8$^+$ T cells, CD80$^+$CD86$^+$ DCs, Treg cells, and M2-TAMs within tumors were gated as CD45$^+$CD8$^+$, CD45$^+$CD8$^+$IFN-γ$^+$, CD45$^+$CD11c$^+$CD80$^+$CD86$^+$, CD45$^+$CD4$^+$CD25$^+$Foxp3$^+$, and CD45$^+$CD11b$^+$F4/80$^+$CD206$^+$ cells, respectively.

For analysis of Tcm cells, the spleens were collected from the mice 1 month after tumor rechallenge and digested into single-cell suspensions. The following antibodies were used: APC/Cy7-conjugated anti-CD45 (BioLegend, 103116), PerCP/Cy5.5-conjugated anti-CD8 (BD, 551162), PE-conjugated anti-CD4 (BioLegend, 100408), BV421-conjugated anti-CD44 (BioLegend, 103040) and PE/Cy7-conjugated anti-CD122 (BioLegend, 123216). The ratios of Tcm cells were analyzed using flow cytometry (BD FACSAria III or Beckman Coulter CytoFlex LX, respectively). The CD4$^+$ Tcm cells and CD8$^+$ Tcm cells were gated as CD45$^+$CD4$^+$CD44$^+$CD122$^+$ cells and CD45$^+$CD8$^+$CD44$^+$CD122$^+$ cells, respectively. For analysis of Ki-67$^+$CD4$^+$ Tcm cells and Ki-67$^+$CD8$^+$ Tcm cells, PerCP/Cy5.5-conjugated anti-Ki-67 (BioLegend, 652423) and PE-conjugated anti-Ki-67 (BioLegend, 652403) were used.

## Cytokine measurement

The secretion of IFN-γ, GZMB, and IL-2 in the serum was assessed by ELISA kits (IFN-γ: Invitrogen, BMS6062; GZMB: Boster, EK1115; IL-2: Boster, EK0398).

## Immunohistochemical and immunofluorescence analyses

The 4T1 tumor sections were stained with anti-E-cadherin (1:500, Servicebio, GB12083), anti-N-cadherin (1:500, Servicebio, GB12135) or anti-Snail (1:400, Servicebio, GB11260) antibodies. The sections were subsequently treated with 3,3'-diaminobenzidine (DAB) (Dako, K5007), counterstained with hematoxylin (Sigma), and examined with a light microscope (Olympus). The percentage area of positive staining for E-cadherin, N-cadherin, and Snail was quantified using the ImageJ software. The CT26 tumor sections were fixed with neutral formalin solution (Servicebio, G1101), stained with anti-E-cadherin (1:500, Servicebio, GB12083) or anti-N-cadherin (1:500, Servicebio, GB12135), and imaged using CLSM.

## Liver, kidney, and heart function examination

The blood samples were centrifuged at 2000×g for 20 min and the serum was collected to measure lactate dehydrogenase (LDH, Nanjing Jiancheng, A020-2-2), creatine kinase-MB (CK-MB, Cusabio, CSB-E14404m), alanine transaminase (ALT, Nanjing Jiancheng, C00921), aspartate transaminase (AST, Nanjing Jiancheng, C01021), creatinine (Cre, Geruisi-bio, G1204W), and blood urea nitrogen (BUN, Geruisi-bio, G1201W) levels according to the manufacturer's instructions.

## Statistical analysis

Sample sizes were chosen based on prior published studies and commonly accepted practices in the field. All results were presented as mean ± SD from at least three independent experiments. The sample sizes are as follows: quantification of R-MHA uptake in 4T1 cells ($n = 10$ independent samples); quantification of Asn levels in 4T1 cells ($n = 6$ independent samples); quantification of ALT, AST, Cre and BUN levels in serum ($n = 3$ independent samples); average tumor growth curves and animal survival curves ($n = 6$ mice); and ratios of CD45$^+$CD4$^+$CD44$^+$CD122$^+$ cells and CD45$^+$CD8$^+$CD44$^+$CD122$^+$ cells ($n = 3$ independent samples). Statistical significance was determined using unpaired two-tailed Student's *t* test, one-way ANOVA with Tukey's post-hoc test, two-way ANOVA with Tukey's multiple comparisons test, or the log-rank (Mantel–Cox) test. Data analysis was conducted using GraphPad Prism 8.0 and OriginPro 2021. All samples/mice were randomly allocated into experimental groups. No data were excluded from the analyses.

## Reporting summary

Further information on research design is available in the Nature Portfolio Reporting Summary linked to this article.

# Data availability

The authors declare that all the data supporting the findings of this study are available within the article, supplementary information or source data file. Source data are provided with this paper.

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

## Acknowledgements

The authors acknowledge the financial support from the National Key R&D Program of China (2024YFA0919300, H.Y.S.), the National Natural Science Foundation of China (32471436 and 32171371, H.Y.S.), the Natural Science Foundation of Shanghai (22ZR1435800, X.Y.J.), Shanghai Municipal Health Commission (22YQ080, X.Y.J.), and China Post-doctoral Science Foundation (2023M742307 and 2024T170569, D.X.G.).

## Author contributions

X.J., H.S., and N.C. conceived the study and designed the experiments. Y.S., H.W., D.G. and J.L. performed the experiments. Y.S. and J.S. analyzed the data. X.J. and H.S. supervised the project. Y.S., X.J. and H.S. wrote the manuscript. All authors discussed the results and have given approval to the final version of the paper.

## Competing interests

The authors declare no competing interests.
