## [Transparent Peer Review file · Nature Communications]

Dual Asparagine-Depriving Nanoparticles Against Solid Tumors

Corresponding Author: Professor Haiyun Song

Version 0:

Reviewer comments:

Reviewer #1

(Remarks to the Author)

This study reports dual-responsive nanoparticles (loaded with Asparaginase/ASP plus Rotenone/Rot) with additional chemical properties that may enable pH/ROS-dependent and tumor-targeted release of two agents. While ASP depletes extracellular asparagine/Asn, Rot inhibits mitochondria complex I activities, which were suggested to be indirectly connected to de novo Asn biosynthesis from other studies. As a result, the intracellular Asn pool would be significantly diminished, leading to cell cycle arrest and cell death. The significance and merits of the current study are based on 1) newly designed NPs enable pH/ROS-responsive and tumor-targeted release of two agents: ASP will be released into extracellular compartment/TME and Rot will be released into the cellular compartment, respectively; 2) the pharmacological properties of the dual-responsive NPs can ensure tumor on-target effects while keeping manageable off-target effects on vital organs/cell types. Depleting Asn (or any of the 20 amino acids) would severely block protein synthesis in any cell type. Impressive efficacies of this novel agent were observed in several preclinical syngeneic tumor models without notable toxicity in the liver and kidney. Another appealing and clinically relevant observation is such an agent seems to induce a T-cell-mediated memory response. The current study's data may have broad implications for developing novel and effective targeted therapy for solid tumors. As such, Asn-depleting therapy can be extended from treating ASNS-silenced leukemia to any tumor type. In addition, a similar strategy may be applied to other chemically compatible agents to achieve local delivery of multiple agents for combinational therapy.

That being said, I have a couple of suggestions for authors to consider:

1) The authors' effort in addressing how dual-responsive NP impacts the immune system is appreciated and encouraged; however, it seems superficial and insufficient. While extensively and broadly assessing the functional impact of dual-responsive NP on all immune subsets is understandably beyond the scope of the current work, superficially profiling immune cell phenotypes failed to provide any meaningful and relevant information. The immunological aspect is one of the highlights of the current study. It should be strengthened with experiments designed to rigorously test functionally relevant immune subsets, effector T cells, Treg, etc, at early time points followed by R-MAHA treatment. In addition, pertinent background information has not been introduced and discussed to justify the experiment and interpretation. For example, Asn metabolism in T cells and its therapeutic implications in the context of immunotherapy have been recently studied. In one study (PMID: 33420490), restricting extracellular Asn suppresses T cell-mediated anti-tumor response, while in another study (PMID: 37550596), an opposite effect was reported. The data (although preliminary) presented in the current study seem to be in line with the latter. Authors need to provide more evidence and discussion to clarify their position.

2) CD44 is highly expressed on the surface of active T cells (considered a T cell activation marker). As such, HA-modified NPs are likely targeting effector T cells (the most crucial component mediating the antitumoral immune response). Figure 3 included CD8 T cell data, which lacks critical information, and the T cell used in the study seems to be naive rather than effector CD8 T cells (active and proliferating). Additional studies are required to address the effects of R-MAHA on active and proliferating effector T cells.

3) Rotenone is a well-known toxic compound. While the dual-responsive NP is designed to be pH-responsive and tumor-targeting, more rigorous studies are required to mitigate the concerns about the potential toxicity of Rot. Inhibiting complex I activity critically impacts the function of several vital organs, particularly cardiac function. Cardiac function tests and cardiac injury biomarkers should be considered in animals treated with dual-responsive NPs (loaded with ASP+Rot)

Minor points:

- 1) Critical technical information is often missing and inconsistently presented in the figure legend and material method section: for example, in Figure 3, there is no further information about BMDM (differentiated or not) and CD8 T cells (naive or activated); timepoints were provided in some but not other experiments; Figure 4e: at which timepoints, tumor cells were isolated to assess Asn level?; In other figures and methods, there is no information about mouse strains used for tumor hosts; similar omissions can be found throughout the manuscript and compromise the scientific rigor of the study.
- 2) The authors claim that one mM H₂O₂ in vitro mimics the ROS levels in the TME without providing any reference or evidence. This needs to be addressed.
- 3) The level of asparagine in plasma and Tumor interstitial fluid (TIF) with or without R-MAHA treatment would provide critical information to assess the pharmacological utility of R-MAHA
- 4) Related to the point above, the observed therapeutic effects are not necessarily due to a proposed ROS-responsive release of ASP to the TME and consequent depletion of Asn in the TEM. It is conceivable that NPs loaded with ASP can be taken up and subsequently released into the intracellular compartment, synergizing with Rot to deplete the intracellular Asn pool. Additional discussion would be necessary if no available experimental evidence can address this point.

Reviewer #2

(Remarks to the Author)

This manuscript by Shen and Wang et al. examines a therapeutic strategy to target solid tumor asparagine dependence. The manuscript builds on previously published findings which showed that combining asparaginase and ETC inhibitors can reduce solid tumor growth. In this manuscript, the authors construct a nanoparticle to co-deliver asparaginase and the ETC inhibitor rotenone specifically to tumor cells in mice. The nanoparticle (R-MAHA) exhibits tumor cell-targeting (with hyaluronidase functionalization), ROS-dependent asparaginase release (to promote release in the ROS-abundant microenvironment), and pH-dependent rotenone release (to release rotenone intracellularly in acidic endosomes). The authors confirm the release of asparaginase and rotenone in vitro using tumor ROS and pH mimicking conditions. They also confirm in vitro R-MHA internalization by endosomal colocalization. The authors then use TNBC and CRC mouse models to show R-MAHA efficacy in vivo. They confirm in vivo tumor targeting and specificity with a fluorescently-labeled R-MAHA and show that unlike free rotenone, rotenone loaded into R-MAHAs does not cause toxicity (liver function, body weight) due to tumor-specific targeting. Moreover, the manuscript shows that R-MAHA administration is significantly and impressively more effective at reducing tumor asparagine levels, tumor growth, and EMT than the combination of free asparaginase and free rotenone. Remarkably, mice that have been healed by R2x-MAHA and finished with treatment are resistant to TNBC and CRC re-challenge, with a concomitant induction of CD4⁺ and CD8⁺ T cells relative to treatment-naïve mice, indicating long-term T cell memory.

This is a thorough manuscript with impressive results. The nanoparticle design allows for tumor cell specificity - sparing immune cells to enhance tumor elimination, removing toxicity, and enabling the use of more effective (but otherwise toxic) drugs in vivo. Prior to publication, the following points should be addressed:

1. An important facet of the R-MAHA nanoparticle is its tumor cell specificity. The authors show that the nanoparticle does not target T cells. However, have the authors measured immune cell infiltration into the tumor environment with R-MAHA in comparison to free asparaginase/rotenone? It would be interesting to know whether capacitating immune cells contributes to the efficacy of R-MAHA.
2. The release of rotenone from R-MAHA is pH-dependent. Since the tumor microenvironment is acidic, does any of the rotenone get released extracellularly, or does exclusively get released in endosomes?
3. The authors showed that asparagine levels are reduced in the tumor after treatment with R-MAHA. Are TCA metabolites and aspartate also reduced with R-MAHA and R-MHA? Are other amino acids unaffected? This information would show that the nanoparticle is functioning as expected.
4. Is there any in vivo validation that asparaginase gets released from R-MAHA extracellularly in the tumor microenvironment?

Minor points:

1. What is the quantitative scale on the heatmaps?

2. The authors state on line 149: "The MHAs loaded with Rot (10-60 ng/mL) exhibited dose-dependent inhibitory effects on the viability of 4T1 cells. In contrast, the same doses of R-MHAs did not cause lethality in BMDMs or CD8+ T cells, manifesting that tumor cells were more sensitive to the inhibition of Asn synthesis than other cell types." Is the cancer cell specific sensitivity because other cell types are not sensitive to asparagine synthesis or because the MHAs target cancer cells with HA and do not get internalized in other cell types.

Reviewer #3

(Remarks to the Author)

This manuscript presents an innovative study on dual Asn-depriving nanoparticles for the treatment of solid tumors, including triple-negative breast cancer (TNBC) and colorectal cancer (CRC). The authors propose a strategy that targets both extracellular and intracellular Asn, overcoming the limitations of current ASNase-based therapies. The study is well-designed, the experiments are comprehensive, and the data convincingly demonstrate the potential of these nanoparticles to inhibit tumor growth, recurrence, metastasis, and induce long-term immune memory. However, there are areas where further clarification, additional data, and methodological refinements would strengthen the conclusions and broaden the applicability of this work.

Major points:

1. Limited Tumor Models: The study focuses on two tumor models, TNBC and CRC, which limits the evidence for the broad applicability of the dual Asn-depleting therapy. Including additional tumor cell lines in vitro, such as pancreatic or lung cancer models, could better demonstrate the nanoparticles' potential across different tumor types. Alternatively, discussing this limitation in the manuscript would strengthen its transparency and highlight directions for future research.

2. Extracellular Asn Depletion: The study clearly demonstrates the dual roles of ASNase and Rot in regulating Asn levels both in vitro and in vivo. However, the impact of ASNase on depleting extracellular Asn in the tumor microenvironment (TME) is not directly addressed. To strengthen the evidence for the dual regulatory mechanism, it would be valuable to measure Asn concentrations in the TME, using methods such as LC-MS or HPLC. This would provide direct validation of extracellular Asn depletion and its contribution to the therapeutic efficacy of the nanoparticles.

3. Immune Cell Analysis in the TME: The article primarily focuses on the response of tumor cells to Asn regulation, while other critical components of the tumor microenvironment (TME), such as immune cells, are not sufficiently studied. Investigating the state of immune cells in the TME, such as their infiltration, activation, and functional status, could provide a more comprehensive understanding of how Asn depletion impacts the TME and uncover its anti-tumor mechanisms.

4. Functional Evidence of Long-Term Memory: In Fig5, the study demonstrates an increase in the proportion of Tcm cells (CD4+ and CD8+) in cured mice, additional functional analyses would strengthen the evidence for long-term memory. Measuring cytokine secretion (e.g., IFN- γ , IL-2) from Tcm cells could confirm their functionality. Furthermore, proliferation assays using markers such as BrdU or Ki-67 would help assess the potential of Tcm cells to expand and maintain the memory response over time. These analyses would provide more direct evidence for the establishment and persistence of long-term immune memory. In addition, the previous work on the role of Asn on T memory cells should be described in the Introduction.

Other points:

1. Cellular uptake of the NPs should be quantified by flowcytometry rather than exclusively relying on imaging quantification

2. Markers for subcellular organelles should be co-stained

3. Different doses of R-M should be examined in Fig. 3c

4. There seems to be signals in the brains in Fig. 4a. If this is true, there will be implications for brain-related toxicity.

5. Define R2x-MAHAs: Twice of contents or NPs?

6. Does the treatment affect Asn levels in plasma and other normal tissues, particularly brains?

Reviewer #4

(Remarks to the Author)

Version 1:

Reviewer comments:

Reviewer #1

(Remarks to the Author)

The authors have rigorously addressed comments raised by me and the other two reviewers. I want to congratulate the authors for establishing a new paradigm, which will undoubtedly provide tremendous value for advancing metabolism-targeting therapies in cancer.

Reviewer #2

(Remarks to the Author)

All of my concerns have been addressed adequately.

Reviewer #3

(Remarks to the Author)

the revision is satisfactory.

Reviewer #4

(Remarks to the Author)

Dear reviewers,

We really appreciate your highly valuable comments, which are of great help to improve the quality of our manuscript. According to the feedbacks, we have carefully revised the manuscript with the newly added data and interpretation, and resubmit it for your evaluation. Additionally, our point-by-point responses are provided as follows.

Point-by-Point Responses

Reviewer #1 (Asparagine depletion, immune cell metabolism):

(1) The authors' effort in addressing how dual-responsive NP impacts the immune system is appreciated and encouraged; however, it seems superficial and insufficient. While extensively and broadly assessing the functional impact of dual-responsive NP on all immune subsets is understandably beyond the scope of the current work, superficially profiling immune cell phenotypes failed to provide any meaningful and relevant information. The immunological aspect is one of the highlights of the current study. It should be strengthened with experiments designed to rigorously test functionally relevant immune subsets, effector T cells, Treg, etc, at early time points followed by R-MAHA treatment. In addition, pertinent background information has not been introduced and discussed to justify the experiment and interpretation. For example, Asn metabolism in T cells and its therapeutic implications in the context of immunotherapy have been recently studied. In one study (PMID: 33420490), restricting extracellular Asn suppresses T cell-mediated anti-tumor response, while in another study (PMID: 37550596), an opposite effect was reported. The data (although preliminary) presented in the current study seem to be in line with the latter. Authors need to provide more evidence and discussion to clarify their position.

Following the reviewer's valuable suggestion, we explored the immunomodulatory effects of these NPs in 4T1 tumors, including the levels of CD8⁺ T cells, the expression of IFN- γ in CD8⁺ T cells, the serum levels of GZMB, and the levels of DC maturation markers, Treg cells, and M2-TAMs. We observed that the R-MAHA treatment significantly promoted CD8⁺ T cell infiltration (Supplementary Figure 23a), upregulated the expression of IFN- γ in CD8⁺ T cells (Supplementary Figure 23b), and stimulated their secretion of GZMB (Supplementary Figure 23c). These effects were further augmented via R^{2x}-MAHAs. These lines of evidence indicate that the NP-mediated Asn restriction in the TME can promote anti-tumor CD8⁺ T cell immunity. This is consistent with the findings of Wang *et al.* and a recent study, both of which indicate that prolonged Asn restriction can potentiate T cell anti-tumor activity (*Nat. Metab.* 2023, 5:1423-1439; *Nat. Metab.* 2025, doi: 10.1038/s42255-025-01245-6). To emphasize this result, we added background information in the 2nd paragraph of "Introduction" and our interpretation in the 1st paragraph of "Discussion". In addition, the analysis of DC maturation markers revealed an upregulated presenting of CD80 and CD86 molecules on the DC surfaces after R-MAHA or R^{2x}-MAHA treatment (Supplementary Figure 24). Furthermore, we analyzed immunosuppressive cell populations, and the data demonstrated that the treatment with R-MAHAs or R^{2x}-MAHAs resulted a significant decrease of Treg cells and a mild reduction in the M2-TAMs (Supplementary Figure 25). Together, these results indicated that the dual Asn-depriving NPs could promote anti-tumor immune responses, which also contributed to their therapeutic efficacy.

Levels of tumor-infiltrating CD8⁺ T cells, IFN- γ ⁺CD8⁺ T cells and serum levels of GZMB:

Supplementary Figure 23. Effects of dual Asn-depriving NPs on tumor-infiltrating CD8⁺ T cells. (a) Flow cytometric analysis (left) and the ratios (right) of CD8⁺ T cells (gated on CD45⁺ population) after indicated treatment. (b) Flow cytometric analysis (left) and the ratios (right) of IFN- γ ⁺CD8⁺ T cells (gated on CD45⁺CD8⁺ population) after indicated treatment. (c) Serum levels of GZMB in tumor-bearing mice after indicated treatment. Data are represented as mean \pm SD (n = 3). One-way ANOVA with Tukey's post-hoc test, * P < 0.05, *** P < 0.001, ns means not significant.

Levels of CD80⁺CD86⁺ DCs within 4T1 tumors:

Supplementary Figure 24. Effects of dual Asn-depriving NPs on DC maturation. Flow cytometric analysis (left) and the ratios (right) of CD80⁺CD86⁺ DCs (gated on CD45⁺CD11c⁺ population) after indicated treatment. Data are represented as mean ± SD (n = 3). One-way ANOVA with Tukey's post-hoc test, ****P* < 0.001, ns means not significant.

Levels of Treg cells and M2-TAMs within 4T1 tumors:

Supplementary Figure 25. Effects of dual Asn-depriving NPs on tumor-infiltrating Treg cells and M2-TAMs. (a) Flow cytometric analysis (left) and the ratios (right) of Treg cells (gated on CD45⁺CD4⁺ population) after indicated treatment. (b) Flow cytometric analysis (left) and the ratios (right) of M2-TAMs (gated on CD45⁺CD11b⁺F4/80⁺ population) after indicated treatment. Data are represented as mean ± SD (n = 3). One-way ANOVA with Tukey's post-hoc test, **P* < 0.05, ****P* < 0.001, ns means not significant.

(2) CD44 is highly expressed on the surface of active T cells (considered a T cell activation marker). As such, HA-modified NPs are likely targeting effector T cells (the most crucial component mediating the antitumoral immune response). Figure 3 included CD8 T cell data, which lacks critical information, and the T cell used in the study seems to be naive rather than effector CD8 T cells (active and proliferating). Additional studies are required to address the effects of R-MAHA on active and proliferating effector T cells.

We thank the reviewer for this question. In the previous manuscript, CD8⁺ T cells used for *in-vitro* studies were naïve cells, and we have changed “CD8⁺ T cell” to “naïve CD8⁺ T cell” in the revised figures (Figure 3a, c; Supplementary Figures 6 and 7, presented as Supplementary Figures 7 and 10 in revised version) and the corresponding text. In the revised manuscript, activated CD8⁺ T cells were also used as controls to investigate the *in-vitro* effects of R-MHAs (*CD44* expression level, tumor cell-targeting capacity, and cytotoxicity). The expression level of *CD44* in 4T1 breast cancer cells was substantially higher than that in undifferentiated bone marrow-derived macrophages (M0-BMDMs), naïve CD8⁺ T cells, and activated CD8⁺ T cells (Supplementary Figure 6). We also used FITC-labeled R-Ms and R-MHAs to examine their internalization by 4T1 tumor cells and activated CD8⁺ T cells *in vitro*. Compared to unmodified micelles, HA functionalization on R-M^{FITC}HAs significantly enhanced cellular internalization of the micelles by 4T1 tumor cells far more than by activated CD8⁺ T cells (Supplementary Figures 7 and 8). Moreover, the tumor-infiltrating CD8⁺ T cell population only represents a small fraction in the tumor, while the number of tumor cells is overwhelmingly dominant. Consequently, the higher levels of *CD44* on tumor cells and their overwhelming abundance make it more likely for R-MAHAs to target tumor cells rather than tumor-infiltrating CD8⁺ T cells in the TME.

We also examined the effect of R-MHAs on the viabilities of 4T1 tumor cells and activated CD8⁺ T cells. The MHAs loaded with Rot (10-70 ng/mL) exhibited severe and dose-dependent inhibitory effects on the viability of 4T1 tumor cells (Supplementary Figure 10a). R-MHAs at the concentrations of 10-50 ng/mL did not cause lethality in activated CD8⁺ T cells, and higher concentrations (60-70 ng/mL) mildly weakened the cell viability, indicating that activated CD8⁺ T cells were less sensitive to Asn synthesis inhibition than tumor cells (Supplementary Figure 10g).

Expression levels of *CD44* in different cells:

Supplementary Figure 6. The levels of *CD44* expression in different types of cells. Data are represented as mean \pm SD (n = 3). Student's *t*-test, ** $P < 0.01$, *** $P < 0.001$, ns means not significant.

Cellular uptake of FITC-labeled R-MHAs:

Supplementary Figure 7. Cellular uptake of FITC-labeled NPs. (a) Confocal fluorescence imaging of intracellular levels of R-micelles^{FITC} (R-M^{FITC}) or R-M^{FITC}HAs in 4T1 cells, M0-BMDMs, naïve CD8⁺ T cells, and activated CD8⁺ T cells at various time points. Scale

bars: 20 μm . (b) Quantifications of intracellular levels of FITC-labeled NPs. Data are represented as mean \pm SD (n = 10). Student's *t*-test, * $P < 0.05$, ** $P < 0.01$, *** $P < 0.001$, ns means not significant.

Supplementary Figure 8. Flow cytometric analysis of cellular uptake of FITC-labeled NPs. (a) Representative flow cytometric plots showing the uptake of R-M^{FITC} or R-M^{FITC}HAs by 4T1 cells, M0-BMDMs, naïve CD8⁺ T cells, and activated CD8⁺ T cells at various time points. (b) Quantifications of intracellular levels of FITC-labeled NPs. Data are represented as mean \pm SD (n = 6). Student's *t*-test, *** $P < 0.001$, ns means not significant.

Cytotoxicity of R-MHAs on 4T1 tumor cells and activated CD8⁺ T cells:

Supplementary Figure 10a, g. Viabilities of 4T1 tumor cells (a) and activated CD8⁺ T cells (g) after 48 hours of indicated treatment. Data are represented as mean \pm SD (n = 3). One-way ANOVA with Tukey's post-hoc test, *** P < 0.001, ns means not significant.

(3) Rotenone is a well-known toxic compound. While the dual-responsive NP is designed to be pH-responsive and tumor-targeting, more rigorous studies are required to mitigate the concerns about the potential toxicity of Rot. Inhibiting complex I activity critically impacts the function of several vital organs, particularly cardiac function. Cardiac function tests and cardiac injury biomarkers should be considered in animals treated with dual-responsive NPs (loaded with ASP + Rot).

Many thanks for raising this concern. In the previous manuscript, we utilized Cy5.5-incorporated M^{Cy5.5}AHAs to study their *in vivo* distribution (Supplementary Figure 10, presented as Supplementary Figure 14 in revised manuscript). *Ex vivo* fluorescence imaging revealed that there was no detectable signal in the heart, indicating minimal NP accumulation. In the revised manuscript, we also performed cardiac function tests and examined the cardiac injury biomarkers (echocardiographic analysis as well as the serum levels of LDH and CK-MB) (Supplementary Figure 19). These results demonstrated that R^{2x}-MAHA treatment did not cause fluctuation of cardiac function parameters.

Supplementary Figure 14. Tumor targeting capacity of HA-decorated NPs. (a) *Ex vivo* fluorescence imaging of tumors and major organs 24 hours after intravenous injection of M^{Cy5.5}As or M^{Cy5.5}AHAs at a dose of 1 mg/kg Cy5.5. (b) Quantification of fluorescence intensities in tumors and major organs. Data are represented as mean \pm SD (n = 3). Student's *t*-test, **P* < 0.05, ***P* < 0.01, ****P* < 0.001, ns means not significant. Abbreviations: tumor, Tu; heart, He; liver, Li; spleen, Sp; lung, Lu; kidney, Ki.

Supplementary Figure 19. Cardiac functional tests. (a) Representative M-mode echocardiography images of tumor-bearing mice after indicated treatment. (b, c) Echocardiographic analysis of ejection fraction and fractional shortening. (d, e) Serum LDH and CK-MB activities in tumor-bearing mice after indicated treatment. Data are represented as mean \pm SD (n = 3). Student's *t*-test, ns means not significant.

Minor points:

(1) Critical technical information is often missing and inconsistently presented in the figure legend and material method section: for example, in Figure 3, there is no further information about BMDM (differentiated or not) and CD8 T cells (naive or activated); timepoints were provided in some but not other experiments; Figure 4e: at which timepoints, tumor cells were isolated to assess Asn level?; In other figures and methods, there is no information about mouse strains used for tumor hosts; similar omissions can be found throughout the manuscript and compromise the scientific rigor of the study.

We feel sorry for the omission of these details. We have revised our description and added the detailed information in revised manuscript. Specifically, the BMDMs used for *in-vitro* studies were undifferentiated, and we have revised “BMDM” to “M0-BMDM” in the

revised figures (Figure 3a, c; Supplementary Figures 6 and 7, presented as Supplementary Figures 7 and 10 in revised version) and in related figure legends. The CD8⁺ T cells used in the previous manuscript were naïve, and we have changed “CD8⁺ T cell” to “naïve CD8⁺ T cell” in the revised figures (Figure 3a, c; Supplementary Figures 6 and 7, presented as Supplementary Figures 7 and 10 in revised version) and in related figure legends. The timepoints for analysis of CD8⁺ T cells, DCs, Treg cells, M2-TAMs, and Tcm cells were provided in the revised Methods (“Immune memory cell analysis”, presented as “Immune cell analysis” in revised version). The timepoints for analyzing Asn levels in isolated tumor cells were provided in the revised figure legends (Figure 4e and Figure 6b). The mouse strains used for *in vivo* studies were described in the Methods “Animal tumor models and treatment”.

(2) The authors claim that one mM H₂O₂ in vitro mimics the ROS levels in the TME without providing any reference or evidence. This needs to be addressed.

Thanks for this suggestion. The extracellular H₂O₂ concentration in the TME is estimated to be approximately 200 μM to 1 mM (*Nat. Rev. Mol. Cell Biol.* 2020, 21:363-383; *Redox Biol.* 2017, 11:613-619), and it is very common to use 1 mM H₂O₂ to mimic the ROS level in the TME *in vitro* (*Sci. Transl. Med.* 2018, 10:eaan3682; *Nat. Commun.* 2022, 13:2688; *Angew. Chem. Int. Ed.* 2024, 63:e202403771). We have added the relevant references in our revised manuscript.

(3) The level of asparagine in plasma and Tumor interstitial fluid (TIF) with or without R-MAHA treatment would provide critical information to assess the pharmacological utility of R-MAHA.

In the revised manuscript, we measured the levels of Asn in plasma and TIF after R-MAHA treatment (Supplementary Figure 22a, b). Treatment with R-MAHAs led to a moderate reduction in plasma Asn levels while inducing a more substantial decrease in TIF. In

contrast, ASNase alone or in combination with Rot significantly reduced Asn levels in plasma, but showed less efficacy in TIF. These data indicated that drug delivery via R-MAHAs could greatly improve the efficacy of Asn-regulating drugs in the TME.

Supplementary Figure 22a, b. The Asn levels in plasma (a) and TIF (b) after indicated treatment. Data are represented as mean \pm SD (n = 3). One-way ANOVA with Tukey's post-hoc test, * $P < 0.05$, ** $P < 0.01$, *** $P < 0.001$, ns means not significant.

(4) Related to the point above, the observed therapeutic effects are not necessarily due to a proposed ROS-responsive release of ASP to the TME and consequent depletion of Asn in the TEM. It is conceivable that NPs loaded with ASP can be taken up and subsequently related into the intracellular compartment, synergizing with Rot to deplete the intracellular Asn pool. Additional discussion would be necessary if no available experimental evidence can address this point.

In the revised manuscript, we constructed dual-labeled NPs consisting of FITC-incorporated MHA cores (M^{FITC}HAs) and AlexaFluor 594-conjugated IgG shells (IgG^{AF594}) to monitor core-shell dissociation *in vivo*. Confocal fluorescence imaging of tumor sections demonstrated that M^{FITC}HAs and IgG^{AF594} exhibited low ratios of colocalization after M^{FITC}IgG^{AF594}HA injection, indicating core-shell dissociation (Supplementary Figure 16a). Moreover, flow cytometric analysis revealed a significant increase of M^{FITC}HA signals in the tumor cells, whereas the intracellular IgG^{AF594} signals remained low (Supplementary

Figure 16b, c). These data confirmed the extracellular disintegration of protein shells from core-shell structured NPs in the TME. More importantly, the substantial reduction of Asn levels in the TIF after treatment with R-MAHAs or R²[×]-MAHAs further validated their extracellular Asn-depleting capacity in the TME (Supplementary Figure 22b, shown in our response to minor point 3 of this reviewer). These data verified the ROS-responsive release of ASNase from R-MAHAs and depletion of extracellular Asn in the TME.

Supplementary Figure 16. Intratumor detachment of core-shell structured NPs. (a) Left: Confocal fluorescence imaging of tumor sections after administration of M^{FITC}IgG^{AF594}HA. Scale bar: 15 μm. Right: The colocalization of IgG^{AF594} with the M^{FITC}HA. (b, c) Flow cytometric analysis of M^{FITC}HA and IgG^{AF594} signals in tumor cells (gated on CD45⁻EpCAM⁺ population). Data are represented as mean ± SD (n = 6). One-way ANOVA with Tukey's post-hoc test, ***P < 0.001, ns means not significant.

Reviewer #2 (cancer metabolism):

(1) An important facet of the R-MAHA nanoparticle is its tumor cell specificity. The authors show that the nanoparticle does not target T cells. However, have the authors measured immune cell infiltration into the tumor environment with R-MAHA in comparison to free asparaginase/rotenone? It would be interesting to know whether capacitating immune cells contributes to the efficacy of R-MAHA.

Many thanks for raising this concern. In the revised manuscript, we analyzed the immune cell populations in the 4T1 tumors after treatment (the levels of CD8⁺ T cells, the expression of IFN- γ in CD8⁺ T cells, the serum levels of GZMB, and the levels of DC maturation markers, Treg cells, and M2-TAMs). The results demonstrated that free Rot did not trigger CD8⁺ T cell recruitment, whereas ASNase or ASNase + Rot treatment led to a mild increase in the ratios of tumor-infiltrating CD8⁺ T cells. R-MAHAs promoted CD8⁺ T cell infiltration with a superior efficiency to that of free drug combination (Supplementary Figure 23a). Consistently, the R-MAHA treatment enhanced CD8⁺ T cell activation more potently than other treatments, as evidenced by elevated levels of tumor-infiltrating IFN- γ ⁺CD8⁺ T cells and serum GZMB (Supplementary Figure 23b, c). The above effects were further augmented via R^{2 \times} -MAHAs. Similar results were found in the analysis of DC maturation markers, as indicated by the increased presenting of CD80 and CD86 molecules on the DC surfaces (Supplementary Figure 24). Moreover, the analysis of immunosuppressive cell populations demonstrated that the treatment with R-MAHAs or R^{2 \times} -MAHAs resulted a significant decrease of Treg cells and a mild reduction in the M2-TAMs (Supplementary Figure 25). Together, these results indicated that the dual Asn-depriving NPs could promote anti-tumor immune responses, which also contributed to their therapeutic efficacy.

Levels of tumor-infiltrating CD8⁺ T cells, IFN- γ ⁺CD8⁺ T cells and serum levels of GZMB:

Supplementary Figure 23. Effects of dual Asn-depriving NPs on tumor-infiltrating CD8⁺ T cells. (a) Flow cytometric analysis (left) and the ratios (right) of CD8⁺ T cells (gated on CD45⁺ population) after indicated treatment. (b) Flow cytometric analysis (left) and the ratios (right) of IFN- γ ⁺CD8⁺ T cells (gated on CD45⁺CD8⁺ population) after indicated treatment. (c) Serum levels of GZMB in tumor-bearing mice after indicated treatment. Data are represented as mean \pm SD (n = 3). One-way ANOVA with Tukey's post-hoc test, * P < 0.05, *** P < 0.001, ns means not significant.

Levels of CD80⁺CD86⁺ DCs within 4T1 tumors:

Supplementary Figure 24. Effects of dual Asn-depriving NPs on DC maturation. Flow cytometric analysis (left) and the ratios (right) of CD80⁺CD86⁺ DCs (gated on CD45⁺CD11c⁺ population) after indicated treatment. Data are represented as mean ± SD (n = 3). One-way ANOVA with Tukey's post-hoc test, ****P* < 0.001, ns means not significant.

Levels of Treg cells and M2-TAMs within 4T1 tumors:

Supplementary Figure 25. Effects of dual Asn-depriving NPs on tumor-infiltrating Treg cells and M2-TAMs. (a) Flow cytometric analysis (left) and the ratios (right) of Treg cells (gated on CD45⁺CD4⁺ population) after indicated treatment. (b) Flow cytometric analysis (left) and the ratios (right) of M2-TAMs (gated on CD45⁺CD11b⁺F4/80⁺ population) after indicated treatment. Data are represented as mean ± SD (n = 3). One-way ANOVA with Tukey's post-hoc test, **P* < 0.05, ****P* < 0.001, ns means not significant.

(2) The release of rotenone from R-MAHA is pH-dependent. Since the tumor microenvironment is acidic, does any of the rotenone get released extracellularly, or does exclusively get released in endosomes?

In the previous manuscript, we studied the Rot-releasing kinetics from Rot-loaded micellar cores under different pH conditions mimicking the extracellular TME (pH = 6.7) and

endosome (pH = 5.0). The results demonstrated that the amount of Rot released from micellar cores was kept at basal levels under extracellular-mimicking conditions (pH = 6.7), and was accelerated under endosome-mimicking condition (pH = 5.0) (Figure 2h, i). These data indicated that the efficient release of Rot occurred primarily in the endosomes after NP internalization.

Figure 2h, i. Cumulative release curves of Rot from R-MAHAs at pH 5.0, pH 6.7 and pH 7.4 with (h) or without 1 mM H₂O₂ (i). Data are represented as mean \pm SD (n = 3).

(3) The authors showed that asparagine levels are reduced in the tumor after treatment with R-MAHA. Are TCA metabolites and aspartate also reduced with R-MAHA and R-MHA? Are other amino acids unaffected? This information would show that the nanoparticle is functioning as expected.

Many thanks for raising this concern. In the revised manuscript, we analyzed the TCA cycle metabolites (fumarate, α -KG, and NAD⁺/NADH), aspartate, glutamate, and methionine in 4T1 tumors after treatment with R^{2x}-MHAs and R^{2x}-MAHAs (Supplementary Figure 21). The results demonstrated that the cellular pools of fumarate, α -KG, aspartate, and the NAD⁺/NADH ratio were shrunken after R^{2x}-MAHA treatment. In the meantime, R^{2x}-MAHAs did not affect the cytosolic glutamate or methionine levels. Notably, similar effects were observed in the R^{2x}-MHA treated group.

Supplementary Figure 21. Effects of Rot-loaded NPs on TCA cycle metabolites, aspartate, glutamate, and methionine. Levels of fumarate (a), α -KG (b), NAD^+/NADH (c), aspartate (d), glutamate (e), and methionine (f) in 4T1 tumors after indicated treatment. Data are represented as mean \pm SD ($n = 3$). One-way ANOVA with Tukey's post-hoc test, $**P < 0.01$, $***P < 0.001$, ns means not significant.

(4) *Is there any in vivo validation that asparaginase gets released from R-MAHA extracellularly in the tumor microenvironment?*

In the revised manuscript, we constructed dual-labeled NPs consisting of FITC-incorporated MHA cores ($\text{M}^{\text{FITC}}\text{HAs}$) and AlexaFluor 594-conjugated IgG shells ($\text{IgG}^{\text{AF594}}$) to monitor core-shell dissociation *in vivo*. Confocal fluorescence imaging of tumor sections demonstrated that $\text{M}^{\text{FITC}}\text{HAs}$ and $\text{IgG}^{\text{AF594}}$ exhibited low ratios of colocalization after $\text{M}^{\text{FITC}}\text{IgG}^{\text{AF594}}\text{HA}$ injection, indicating core-shell dissociation (Supplementary Figure 16a). Moreover, flow cytometric analysis revealed a significant increase of $\text{M}^{\text{FITC}}\text{HA}$ signals in the tumor cells, whereas the intracellular $\text{IgG}^{\text{AF594}}$ signals remained low (Supplementary Figure 16b, c). These data validated the extracellular disintegration of protein shells from core-shell structured NPs in the TME.

Supplementary Figure 16. Intratumor detachment of core-shell structured NPs. (a) Left: Confocal fluorescence imaging of tumor sections after administration of M^{FITC}IgG^{AF594}HAs. Right: The colocalization of IgG^{AF594} with the M^{FITC}HAs. Scale bar: 15 μm . (b, c) Flow cytometric analysis of M^{FITC}HA and IgG^{AF594} signals in tumor cells (gated on CD45⁻EpCAM⁺ population). Data are represented as mean \pm SD (n = 6). One-way ANOVA with Tukey's post-hoc test, *** $P < 0.001$, ns means not significant.

Minor points:

(1) *What is the quantitative scale on the heatmaps?*

We are very sorry for the omissions. In the revised manuscript and supporting information, we have added quantitative scales to the heatmaps (i.e., Figure 3e, 4a, 4f, 5b, 5d, 5f, 6c, 6f, 6g; Supplementary Figure 10a, 25, and 28a, presented as Supplementary Figure 14a, 37, and 40a in the revised supporting information).

(2) *The authors state on line 149: "The MHAs loaded with Rot (10-60 ng/mL) exhibited dose-dependent inhibitory effects on the viability of 4T1 cells. In contrast, the same doses*

of R-MHAs did not cause lethality in BMDMs or CD8⁺ T cells, manifesting that tumor cells were more sensitive to the inhibition of Asn synthesis than other cell types.” Is the cancer cell specific sensitivity because other cell types are not sensitive to asparagine synthesis or because the MHAs target cancer cells with HA and do not get internalized in other cell types.

The internalization assay was conducted to compare the speed of NP uptake. It is common to assess internalization shortly after incubation with cells (e.g., ~4 hour or less) (*Nat. Nanotechnol.* 2024, 19:1723; *Nat. Commun.* 2023, 14:1974; *Nat. Commun.* 2023, 14:4771; *Adv. Mater.* 2022, 34:e2207593), as NPs without targeting molecules can still be internalized by cultured cells after longer incubation. To assess the tumor cell-targeting capacity of the micellar surface-decorated HA, we analyzed the results in the time window of 1-4 hours (Supplementary Figure 6, presented as Supplementary Figure 7 in the revised manuscript). Compared to unmodified micelles, HA functionalization on the R-MHAs significantly enhanced cellular internalization of the micelles by 4T1 tumor cells rather than M0-BMDMs or naïve CD8⁺ T cells, and far more than by activated CD8⁺ T cells, validating the tumor cell-targeting capacity of the micellar surface-decorated HA. Notably, after prolonged incubation (24 hours), unmodified and HA-modified micelles exhibited same levels of internalization in 4T1 cells, M0-BMDMs, naïve CD8⁺ T cells, or activated CD8⁺ T cells (data for reviewers only). In the cytotoxicity assay, cell viability was measured 48 hours after R-MHA treatment. Collectively, we attribute the cancer cell-specific sensitivity of R-MHAs primarily to the higher dependence of tumor cells on Asn synthesis, rather than enhanced internalization facilitated by their tumor cell-targeting capacity. This observation is consistent with previous reports demonstrating that tumor cells have an increased demand for Asn to support rapid proliferation (*Nat. Commun.* 2016, 7:11457; *Nat. Cancer* 2022, 3:1386-1403; *Cancer Res.* 2024, 84:3004-3022).

Supplementary Figure 7. Cellular uptake of FITC-labeled NPs. (a) Confocal fluorescence imaging of intracellular levels of R-micelles^{FITC} (R-M^{FITC}) or R-M^{FITC}HAs in 4T1 cells, M0-BMDMs, naïve CD8⁺ T cells, and activated CD8⁺ T cells at various time points. Scale bars: 20 μ m. (b) Quantifications of intracellular levels of FITC-labeled NPs. Data are represented as mean \pm SD (n = 10). Student's *t*-test, **P* < 0.05, ***P* < 0.01, ****P* < 0.001, ns means not significant.

Data for reviewers only. Confocal fluorescence imaging and quantifications of intracellular levels of R-micelles^{FITC} (R-M^{FITC}) or R-M^{FITC}HAs in cultured 4T1 tumor cells, M0-BMDMs, naïve CD8⁺ T cells, and activated CD8⁺ T cells after 24-hour incubation. Scale bars: 15 μ m. Data are represented as mean \pm SD (n = 10). Student's *t*-test, ns means not significant.

Reviewer #3 (preclinical cancer models, CRC):

Major points:

(1) Limited Tumor Models: The study focuses on two tumor models, TNBC and CRC, which limits the evidence for the broad applicability of the dual Asn-depleting therapy. Including additional tumor cell lines in vitro, such as pancreatic or lung cancer models, could better demonstrate the nanoparticles' potential across different tumor types. Alternatively, discussing this limitation in the manuscript would strengthen its transparency and highlight directions for future research.

Thanks for this suggestion. In the revised manuscript, the applicability of dual Asn-depleting therapy via R-MAHAs was also tested in Panc02 pancreatic cancer cells, LLC lung cancer cells, and H22 hepatic cancer cells cultured in standard or Asn-supplemented medium. Specifically, we measured intracellular Asn levels (Supplementary Figure 12a-c) and cell proliferation rates (Supplementary Figure 12d-f) in these cell lines after incubation with H₂O₂ pre-treated R-MAHAs. These data demonstrated that R-MAHAs could efficiently induce dual Asn deprivation and suppress cell proliferation in Panc02, LLC, and H22 cell lines.

Supplementary Figure 12. Intracellular levels of Asn (a-c) and relative proliferation rates (d-f) of Panc02 cells, LLC cells, and H22 cells after the indicated treatment, with or without supplementing Asn (0.1 mM) in the cell culture medium. R-MHAs (Rot: 50 ng/mL) and R-MAHAs (Rot: 50 ng/mL; ASNase: 0.6 μ g/mL) were used. Data are represented as mean \pm SD (n = 6). One-way ANOVA with Tukey's post-hoc test, * P < 0.05, ** P < 0.01, *** P < 0.001, ns means not significant.

(2) *Extracellular Asn Depletion: The study clearly demonstrates the dual roles of ASNase*

and Rot in regulating Asn levels both *in vitro* and *in vivo*. However, the impact of ASNase on depleting extracellular Asn in the tumor microenvironment (TME) is not directly addressed. To strengthen the evidence for the dual regulatory mechanism, it would be valuable to measure Asn concentrations in the TME, using methods such as LC-MS or HPLC. This would provide direct validation of extracellular Asn depletion and its contribution to the therapeutic efficacy of the nanoparticles.

In the revised manuscript, we isolated the tumor interstitial fluid (TIF) and directly measured extracellular Asn levels in the TME (Supplementary Figure 22b). The results revealed a significant reduction of Asn levels in the TIF after treatment with R-MAHAs or R²×-MAHAs, validating their capacities to deplete extracellular Asn in the TME.

Supplementary Figure 22b. The Asn levels in TIF after indicated treatment. Data are represented as mean \pm SD (n = 3). One-way ANOVA with Tukey's post-hoc test, ** $P < 0.01$, *** $P < 0.001$, ns means not significant.

(3) *Immune Cell Analysis in the TME:* The article primarily focuses on the response of tumor cells to Asn regulation, while other critical components of the tumor microenvironment (TME), such as immune cells, are not sufficiently studied. Investigating the state of immune cells in the TME, such as their infiltration, activation, and functional status, could provide a more comprehensive understanding of how Asn depletion impacts the TME and uncover its anti-tumor mechanisms.

Many thanks for raising this concern. In the revised manuscript, we analyzed the immune

cell populations in the 4T1 tumors after treatment (the levels of CD8⁺ T cells, the expression of IFN- γ in CD8⁺ T cells, the serum levels of GZMB and the levels of DC maturation markers, Treg cells, and M2-TAMs). The results demonstrated that free Rot did not trigger CD8⁺ T cell recruitment, whereas ASNase or ASNase + Rot treatment led to a mild increase in the ratios of tumor-infiltrating CD8⁺ T cells. R-MAHAs promoted CD8⁺ T cell infiltration with a superior efficiency to that of free drug combination (Supplementary Figure 23a). Consistently, the R-MAHA treatment enhanced CD8⁺ T cell activation more potently than other treatments, as evidenced by elevated levels of tumor-infiltrating IFN- γ ⁺CD8⁺ T cells and serum GZMB (Supplementary Figure 23b, c). The above effects were further augmented via R^{2 \times} -MAHAs. Similar results were found in the analysis of DC maturation markers, as indicated by the increased presenting of CD80 and CD86 molecules on the DC surfaces (Supplementary Figure 24). Moreover, the analysis of immunosuppressive cell populations demonstrated that the treatment with R-MAHAs or R^{2 \times} -MAHAs resulted a significant decrease of Treg cells and a mild reduction in the M2-TAMs (Supplementary Figure 25). Together, these results indicated that the dual Asn-depriving NPs could promote anti-tumor immune responses, which also contributed to their therapeutic efficacy.

Levels of tumor-infiltrating CD8⁺ T cells, IFN- γ ⁺CD8⁺ T cells and serum levels of GZMB:

Supplementary Figure 23. Effects of dual Asn-depriving NPs on tumor-infiltrating CD8⁺ T cells. (a) Flow cytometric analysis (left) and the ratios (right) of CD8⁺ T cells (gated on CD45⁺ population) after indicated treatment. (b) Flow cytometric analysis (left) and the ratios (right) of IFN- γ ⁺CD8⁺ T cells (gated on CD45⁺CD8⁺ population) after indicated treatment. (c) Serum levels of GZMB in tumor-bearing mice after indicated treatment. Data are represented as mean \pm SD (n = 3). One-way ANOVA with Tukey's post-hoc test, **P* < 0.05, ****P* < 0.001, ns means not significant.

Levels of CD80⁺CD86⁺ DCs within 4T1 tumors:

Supplementary Figure 24. Effects of dual Asn-depriving NPs on DC maturation. Flow cytometric analysis (left) and the ratios (right) of CD80⁺CD86⁺ DCs (gated on CD45⁺CD11c⁺ population) after indicated treatment. Data are represented as mean ± SD (n = 3). One-way ANOVA with Tukey's post-hoc test, ****P* < 0.001, ns means not significant.

Levels of Treg cells and M2-TAMs within 4T1 tumors:

Supplementary Figure 25. Effects of dual Asn-depriving NPs on tumor-infiltrating Treg cells and M2-TAMs. (a) Flow cytometric analysis (left) and the ratios (right) of Treg cells (gated on CD45⁺CD4⁺ population) after indicated treatment. (b) Flow cytometric analysis (left) and the ratios (right) of M2-TAMs (gated on CD45⁺CD11b⁺F4/80⁺ population) after indicated treatment. Data are represented as mean ± SD (n = 3). One-way ANOVA with Tukey's post-hoc test, *P < 0.05, ***P < 0.001, ns means not significant.

(4) Functional Evidence of Long-Term Memory: In Fig 5, the study demonstrates an increase in the proportion of Tcm cells (CD4⁺ and CD8⁺) in cured mice, additional functional analyses would strengthen the evidence for long-term memory. Measuring cytokine secretion (e.g., IFN- γ , IL-2) from Tcm cells could confirm their functionality. Furthermore, proliferation assays using markers such as BrdU or Ki-67 would help assess

the potential of Tcm cells to expand and maintain the memory response over time. These analyses would provide more direct evidence for the establishment and persistence of long-term immune memory. In addition, the previous work on the role of Asn on T memory cells should be described in the Introduction.

Many thanks for these questions. Here we provide our responses.

(1) **Concerning the functional evidence of long-term memory.** Apart from analyzing the amount of CD4⁺ and CD8⁺ Tcm cells, we also examined their functions (expression of the cell proliferation marker Ki-67 as well as the secreted levels of IFN- γ and IL-2 in the serum) in the revised manuscript. FACS analysis revealed that both CD4⁺ and CD8⁺ Tcm cells exhibited higher levels of Ki-67 expression in the cured mice compared to the naïve mice, indicating enhanced proliferative capacity (Supplementary Fig. 41a, b). In addition, tumor rechallenge in cured mice strongly stimulated Tcm cell activation, as evidenced by increased IFN- γ and IL-2 secretion in the serum (Supplementary Figs. 33 and 41c, d). These data further verified the induction of long-term memory by dual Asn-depriving NPs in murine models of TNBC and CRC.

Supplementary Figure 33. Tcm cell activation. Serum levels of IFN- γ (a) and IL-2 (b) after 4T1 tumor rechallenge. Data are represented as mean \pm SD (n = 3). Student's *t*-test, ****P* < 0.001, ns means not significant.

Supplementary Figure 41. Tcm cell proliferation and activation in CT26 rechallenged mice. (a, b) Intracellular Ki-67 expression in CD4⁺ Tcm (gated on CD45⁺CD4⁺CD44⁺CD122⁺ population) (a) and CD8⁺ Tcm cells (gated on CD45⁺CD8⁺CD44⁺CD122⁺ population) (b). (c, d) Serum levels of IFN-γ (c) and IL-2 (d) after tumor rechallenge. Data are represented as mean ± SD (n = 5). Student's *t*-test, ****P* < 0.001, ns means not significant.

(2) **Concerning the role of Asn metabolism in induction of T cell memory.** Many thanks for this suggestion. We have added contents about the impact of Asn regulation on T cell memory in the 2nd paragraph of “Introduction”.

Other points:

(1) Cellular uptake of the NPs should be quantified by flow cytometry rather than exclusively relying on imaging quantification.

In the revised manuscript, we compared the cellular uptake of unmodified R-Ms and HA-decorated R-MHAs using flow cytometric analysis (Supplementary Figure 8). The results demonstrated that HA functionalization on R-M^{FITC}HAs significantly enhanced cellular internalization of the micelles by 4T1 tumor cells rather than by M0-BMDMs or naïve CD8⁺ T cells, and far more than by activated CD8⁺ T cells, further validating the tumor cell-targeting capacity of the R-MHAs.

Supplementary Figure 8. Flow cytometric analysis of cellular uptake of FITC-labeled NPs. (a) Representative flow cytometric plots showing the uptake of R-M^{FITC} or R-

M^{FITC}HAs by 4T1 cells, M0-BMDMs, naïve CD8⁺ T cells, and activated CD8⁺ T cells at various time points. (b) Quantifications of intracellular levels of FITC-labeled NPs. Data are represented as mean \pm SD (n = 6). Student's *t*-test, ****P* < 0.001, ns means not significant.

(2) Markers for subcellular organelles should be co-stained.

In the previous manuscript, endosomes were stained to monitor the subcellular localization of R-MHAs (Figure 3b). In the revised manuscript, we also stained mitochondria and endoplasmic reticulum (Supplementary Figure 9a, b) and monitored the colocalization coefficients between R-MHAs and mitochondria, endoplasmic reticulum or endosomes (Supplementary Figure 9c). The results confirmed that the internalized R-MHAs were primarily localized in endosomes.

Figure 3b. Intracellular localization of R-M^{FITC}HAs. Scale bars: 10 μ m.

Supplementary Figure 9. Intracellular localization of R-M^{FITC}HAs. (a, b) Confocal fluorescence imaging of intracellular R-M^{FITC}HAs in 4T1 cells. Mito-Tracker or ER-Tracker was used to label the mitochondria (Mito) or endoplasmic reticulum (ER). Scale bars: 10 μ m. (c) Values of Pearson's correlation coefficient between R-M^{FITC}HAs and Mito, ER, or endosome. Data are represented as mean \pm SD (n = 6). Student's *t*-test, ****P* < 0.001.

(3) Different doses of R-M should be examined in Fig. 3c.

In the revised manuscript, we assessed the cytotoxicity of R-MHAs at different dosages in various types of tumor cells and immune cells, including 4T1 breast cancer cells, Panc02 pancreatic cancer cells, LLC lung cancer cells, H22 hepatic cancer cells, M0-BMDMs, naïve CD8⁺ T cells, and activated CD8⁺ T cells (Supplementary Figure 10). The results showed that R-MHAs loaded with Rot (10-70 ng/mL) exhibited severe and dose-dependent inhibitory effects on the viability of 4T1, Panc02, LLC and H22 cells. In contrast, R-MHAs (10-60 ng/mL Rot) did not cause lethality in M0-BMDMs or naïve CD8⁺ T cells, R-MHAs at 10-50 ng/mL did not affect the cell viability of activated CD8⁺ T cells, and higher

concentrations only mildly weakened the cell viabilities.

Supplementary Figure 10. Cytotoxicity of R-MHAs on tumor cells and immune cells. Viability of 4T1 cells (a), Panc02 cells (b), LLC cells (c), H22 cells (d), M0-BMDMs (e), naïve CD8⁺ T cells (f), and activated CD8⁺ T cells (g) after 48 hours of indicated treatment. Data are represented as mean ± SD (n = 3). One-way ANOVA with Tukey's post-hoc test, *** $P < 0.001$, ns means not significant.

(4) There seems to be signals in the brains in Fig. 4a. If this is true, there will be implications for brain-related toxicity.

Many thanks for raising this concern. To distinguish the signals, we injected free Cy5.5 into healthy mice or 4T1 tumor-bearing mice and performed *in vivo* and *ex vivo* fluorescence imaging (data for reviewers only). The results of *in vivo* fluorescence imaging

demonstrated that fluorescence signals were also observed in the head region of both healthy and tumor-bearing mice after injection of free Cy5.5, indicating that the signals were not due to NP accumulation. More importantly, *ex vivo* fluorescence imaging confirmed that no detectable signals were present in the brain tissues from free Cy5.5- or M^{Cy5.5}AHA-treated groups. Therefore, we believe that the fluorescence observed in the head region in Fig. 4a represents false-positive signals.

Data for reviewers only. *In vivo* and *ex vivo* fluorescence imaging of healthy mice (a) and tumor-bearing mice (b) 24 hours after intravenous injection of free Cy5.5 or M^{Cy5.5}AHAs (Cy5.5: 1 mg/kg).

(5) Define R2x-MAHAs: Twice of contents or NPs?

We are very sorry for this confusion. In our study, R^{2x}-MAHAs contain twice amount of encapsulated Rot compared to R-MAHAs, while maintaining the ASNase and NP mass unchanged. We have added this explanation in the revised context (line 216, page 10).

(6) Does the treatment affect Asn levels in plasma and other normal tissues, particularly brains?

In the revised manuscript, we measured Asn levels in plasma and major organs (brain, heart, liver, spleen, lung, and kidney) after treatment (Supplementary Figure 22). Treatment with R-MAHAs or R^{2×}-MAHAs efficiently reduced Asn levels in TIF and to a lesser extent in serum, but did not affect Asn levels in major organs, including heart, liver, spleen, lung, kidney, and brain. In comparison, ASNase alone or in combination with Rot induced an obvious reduction of Asn levels in plasma and to a lesser degree in TIF, and caused a mild decrease of Asn levels in heart, liver, spleen, and kidney.

Supplementary Figure 22. The Asn levels in plasma (a), TIF (b), and major organs (c-h) after indicated treatment. Data are represented as mean \pm SD (n = 3). One-way ANOVA with Tukey's post-hoc test, * $P < 0.05$, ** $P < 0.01$, *** $P < 0.001$, ns means not significant.